# Towards Croppable Implicit Neural Representations

**Maor Ashkenazi**
Ben-Gurion University of the Negev
maorash@post.bgu.ac.il

**Eran Treister**
Ben-Gurion University of the Negev
erant@cs.bgu.ac.il

## Abstract

Implicit Neural Representations (INRs) have peaked interest in recent years due to their ability to encode natural signals using neural networks. While INRs allow for useful applications such as interpolating new coordinates and signal compression, their black-box nature makes it difficult to modify them post-training. In this paper we explore the idea of editable INRs, and specifically focus on the widely used cropping operation. To this end, we present Local-Global SIRENs – a novel INR architecture that supports cropping by design. Local-Global SIRENs are based on combining local and global feature extraction for signal encoding. What makes their design unique is the ability to effortlessly remove specific portions of an encoded signal, with a proportional weight decrease. This is achieved by eliminating the corresponding weights from the network, without the need for retraining. We further show how this architecture can be used to support the straightforward extension of previously encoded signals. Beyond signal editing, we examine how the Local-Global approach can accelerate training, enhance encoding of various signals, improve downstream performance, and be applied to modern INRs such as INCODE, highlighting its potential and flexibility. Code is available at https://github.com/maorash/Local-Global-INRs.

## 1 Introduction

Neural networks have proven to be an effective tool for learning representations of various natural signals. This advancement offers a paradigm of representing a signal without explicitly defining it. The general idea behind an implicit neural representation (INR) is to model the signal as a prediction task from some coordinate system to the signal values at that coordinate system. This is usually performed using a Multi-Layer Perceptron (MLP), composed of fully connected layers. Once trained, the signal has been implicitly encoded in the weights of the network. These implicit representations have been shown to be useful in various scenarios from interpolating values at new coordinates [29], through signal compression [14] and can be treated as embeddings for downstream tasks [13].

Although INRs exhibit flexibility within their input space and are well-suited for tasks involving high-dimensional data, their black-box nature makes the encoded signal difficult to modify post-training. A trivial strategy involves editing the original signal, followed by training a new INR to encode the modified signal. Another option is fine-tuning the existing INR to encode the modified signal, but it requires preserving the INR size and architecture, which may not be ideal. Other methods attempted to apply direct transformations on the weight space to modify the encoded signal [42, 30].

In this paper, our primary focus is on the fundamental signal editing operation of cropping. We wish to be able to *remove parts of the encoded signal, with a proportional decrease of INR weights*. Although not possible in previous approaches, in principle this should be obtained without any additional fine-tuning, since it conceptually does not require encoding additional information. As a secondary task, we wish to extend an encoded signal effectively. This is not trivial in a simple setting, since encoding additional information may require increased capacity in terms of parameters, and altering the number of parameters makes it difficult to leverage previously encoded information.

38th Conference on Neural Information Processing Systems (NeurIPS 2024).

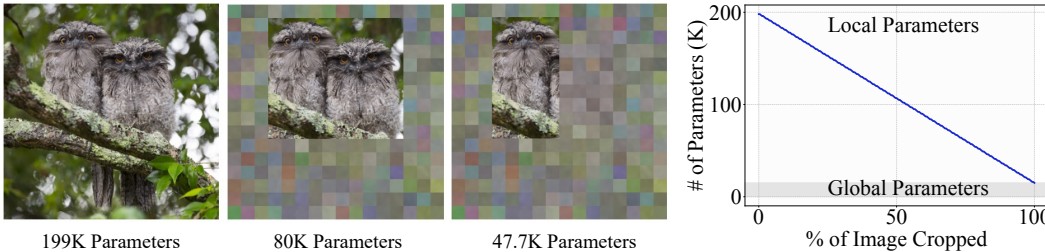

Figure 1: Examples of cropping a Local-Global SIREN with 199k parameters. Plot on the right shows the number of parameters as a function of cropped partitions in the encoded image.

As a natural way to obtain our goals, we first partition the input signal space. The granularity of the partitioning will determine the ability to edit the INR with greater detail. A straightforward approach involves training a compact INR for each partition of the signal. Subsequently, cropping and extending signals are simple. To crop specific partitions, one simply eliminates the INRs corresponding to those partitions. To extend INRs, new signal partitions can be seamlessly incorporated by training additional compact INRs on the new partitions, subsequently adding them to the ensemble. Notably, this straightforward approach offers a significant speed benefit. While requiring a unique implementation, utilizing an ensemble of compact INRs for both encoding and reconstructing the signal is much more efficient than a fully connected INR. Indeed, in KiloNeRF [35], a 3D scene was encoded by adopting the INR-per-Partition approach, leading to notable improvements in rendering times.

However, this partitioning approach lacks a global context, resulting in increased reconstruction error and undesired artifacts. In [35, 15], this was solved using knowledge distillation. Initially, a large INR was trained to encode the entire signal. Next, the encoded representation was distilled into the ensemble of compact INRs. While this approach facilitates rapid rendering, it involves a more intricate training process, demanding additional time to train both the full INR and the compact INRs.

In this work, we present a novel INR architecture based on the premise of combining both local and global context learning. Local features are learned via multiple compact local sub-networks, while global features are learned via a larger sub-network. The features of the local and global networks are interleaved throughout the forward pass, resulting in relatively high reconstruction accuracy, without additional training steps, while also gaining latency improvements due to the local feature learning. While these ideas can be applied to any MLP-based INR, we focus on SIREN [38] for its quality and widespread popularity. We further explore INCODE [20], a state-of-the-art (SOTA) INR, as a potential baseline architecture. We summarize our contribution as follows:

- We propose Local-Global SIRENs, a novel INR architecture that supports cropping parts of the signal with proportional weight decrease, as depicted in Figure 1, all without retraining.
- We analyze Local-Global SIRENs' inherent tradeoff between latency and accuracy by tuning the granularity of input space partitioning.
- We show how Local-Global SIRENs can be leveraged to extend a previously encoded signal.
- We apply the Local-Global architecture on INCODE and solve various downstream tasks, demonstrating the adaptability of our proposed approach.
- We present cases where Local-Global INRs improve upon the baseline INRs.

## 2 Related work

**Implicit neural representations**  INRs have found applications in many diverse tasks such as image super resolution [8, 31], image inpainting [38], zero-shot image denoising [21], image compression [12], image interpolation [4], video encoding [6], camera pose estimation [45], and encoding the weights of neural networks [3]. Another work demonstrating the versatility of INRs, [13] has shown that they can be treated as data points, utilizing their weights as latent vectors for downstream tasks. For 3D shapes, the works of [28, 10, 34] approached the task by formulating a mapping from 3D coordinates to occupancy. Alternatively, [33] has characterized the task as a Signed Distance

Function (SDF). [39] expanded upon this concept by incorporating information related to object appearance. Based on these foundations, [29] predicted the density and color of points within a 5D coordinate system. This approach facilitated the incorporation of high-frequency details through the use of positional encodings. A parallel concept, introduced by [38], utilizes *sine* non-linearities to encapsulate high-frequency details. [20] extended upon this idea, by introducing a harmonizer network tasked with modifying the properties of the nonlinearities. Alternatively, [11, 24] leveraged multiscale representations, gradually incorporating finer features throughout the layers.

**Editing neural representations**    The realm of editing INRs is still in early stages. [42, 32] introduced methods for applying signal processing operators to INRs, involving high-order derivative computation graphs. Alternatively, [30] addressed the challenge of weight space symmetry by proposing an architecture tailored for learning within these spaces. This architecture can later be used to directly edit the weights of a given network. Shifting to 3D scenes, [7] demonstrated scene editing through point cloud manipulation, while [17] introduced a learned color palette for dynamic color changes. Style transfer for INRs was explored by [16], while [43] focused on shape geometry modification, offering techniques for smoothing and deformation.

**Input space partitioning**    Processing local regions by input space partitioning has shown promise in improved capturing of local details. [5, 27] proposed learning a local latent vector per partition. The former passed the latent vector as an input to the INR, while the latter used it as input for a modulator function. Both have shown that when encoding scenes with intricate details, partitioning allows for finer detailed reconstruction. Similar ideas were presented in [40, 19]. [9] used multiple INRs, handling partitions of different resolutions, and [41] proposed an extension by clustering partitions based on shared patterns. Similar clustering methods where leveraged for 3D face modeling [46, 47]. [1] used an INR with sparse heads, corresponding to image partitions, to improve generalization. Partitioning was additionally used to enhance compression capabilities. Specifically, [14] proposed a compression scheme, where a preliminary partitioning step is performed on large scale signals. Similarly, [44] employed a hierarchical tree structure, sharing parameters among similar partitions.

Partitioning has also been used to optimize latency. [37] accelerated training by employing multiscale coarse-to-fine INRs and dynamically selecting regions needing finer details. While compelling, this results in uneven weight distribution throughout partitions of the signal, hindering cropping with relative weight reduction. [25] accelerated convergence by partitioning the input signal space via semantic segmentation and training smaller INRs per segment. [23] accelerated INRs by independently learning each input dimension, which shares a resemblance to partitioning methods. Inspired by long 3D rendering time, [35, 15] offered alternative compact INR-per-partition approaches. As mentioned before, while these methods could apply to our case, we aim to avoid the extra full INR training step.

## 3   Method

### 3.1   Background

In its core, an INR is a mapping function, $F : \mathcal{X} \to \mathcal{Y}$, where $\mathcal{X} \subseteq \mathbb{R}^n$ is the input coordinate space, $\mathcal{Y} \subseteq \mathbb{R}^m$ is the values space, and $F$ is a neural network. An RGB image, for example, corresponds to $\mathcal{X} \subseteq \mathbb{R}^2$ for pixel coordinates and $\mathcal{Y} \subseteq \mathbb{R}^3$ for RGB values. A standard choice for $F$ is an MLP, meaning it is composed of a series of fully connected linear layers, followed by non-linear activation functions. Before being passed to the network, the input coordinates undergo some transformation. In SIREN, the input coordinates are normalized, commonly to $(-1, 1)$, and the activation functions are *sine*. A SIREN layer is defined by $\mathbf{l}_i(x) = \sin\left(\omega \cdot (\mathbf{W}_i \cdot x + \mathbf{b}_i)\right)$, where $\mathbf{W}_i, \mathbf{b}_i$ are the learned parameters of layer $\mathbf{l}_i$ and $\omega$ is a hyperparameter representing the frequency of the *sine* wave.

### 3.2   Partitioning the signal

We begin by partitioning the signal, *as cropping is achieved at the partition level*. We assume the input coordinate space $\mathcal{X}$ can be bounded by an $n$ dimensional hyperrectangle, where the $i$-th dimension has boundaries $\left[B_{\min}^i, B_{\max}^i\right]$. Each dimension is split into $C_i$ equally sized partitions, resulting in $\prod_{i=1}^n C_i$ non-overlapping partitions. The coordinate values are determined before partitioning and remain unaltered thereafter. Each of the partitions is indexed by $n$ coordinates, $(P_0, P_1, \ldots, P_{n-1}) \in \mathbb{N}^n$, where $0 \leq P_i \leq C_i$. This process is demonstrated in Figure 2. A point $p$ with coordinates $(p_0, p_1, \ldots, p_{n-1})$ is mapped to its respective partition by:

$$\text{Partition}(p) = \left( \left\lfloor \frac{p_0 - B_{\min}^0}{\Delta_0} \right\rfloor, \ldots, \left\lfloor \frac{p_{n-1} - B_{\min}^{n-1}}{\Delta_{n-1}} \right\rfloor \right),$$

where $\Delta_i = \frac{B_{\max}^i - B_{\min}^i}{C_i}$ represents the size of each partition in dimension $i$.

### 3.3 Local-Global architecture

Our architecture is composed of *Local Sub-Networks*, each responsible for a specific partition, and a *Global Sub-Network*, familiar with the entire signal. This design facilitates simultaneous extraction of *local features* using the former and *global features* using the latter. The *Merge Operator* is responsible for combining the intermediate local and global features throughout the architecture. Here, for example, we describe our approach with SIREN as the baseline INR, utilizing $sine$ nonlinearities throughout the network. An overview of our proposed architecture is demonstrated in Figure 3.

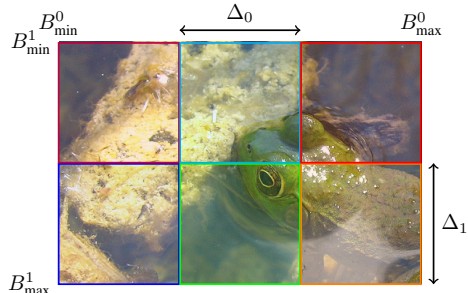

Figure 2: Example of partitioning an image. This trivial example uses $C_0 = 3, C_1 = 2$. To achieve flexible cropping, one must choose larger partition factors.

**Local sub-networks**  We employ multiple local sub-networks, each for a specific partition without shared weights, for a total of $\prod_{i=1}^{n} C_i$ sub-networks. The local sub-networks are composed of SIREN layers, interleaved with the merge operator, described below. The final layer of each local sub-network outputs the partition's values. Given an input point $p$, we pass it on to the local sub-network corresponding to Partition$(p)$. Cropping is natively supported by eliminating entire local sub-networks. While similar to the idea of KiloNeRF, in Local-Global SIRENs we do not require additional pre-training or knowledge distillation steps, significantly reducing overheads.

**Global sub-network**  While the local sub-networks are able to extract details of local patterns, they lack global context, resulting in a subpar reconstruction accuracy compared to SIREN. This is demonstrated qualitatively in Figure 4, and quantitatively throughout our experiments. To remediate this, we utilize an additional SIREN, termed the global sub-network. Its objective is to extract features that are crucial for the global context of the signal. The global features are subsequently combined with the local features through the merge operator. The global sub-network comprises one less layer compared to the local sub-networks; it does not output signal values but is used solely to augment the intermediate local features with global contextual information.

**Merge operator**  The merge operator is performed per coordinate, meaning that a coordinate's local features are only merged with the same coordinate's global features, upholding the definition of an INR. The merge operator consists of (1) a concatenation operation, and (2) a linear layer with a subsequent activation function. The linear layer reduces the concatenated vector's size, to the hidden size of the local sub-networks. It is given by:

$$\text{Merge}(\mathbf{L}, \mathbf{G}) = \sigma(\text{concat}([\mathbf{L}, \mathbf{G}]) \cdot \mathbf{W} + \mathbf{b}) \tag{1}$$

where $\mathbf{L}$ and $\mathbf{G}$ represent the local and global features, respectively. The term $\sigma$ denotes the activation function, in our case, $sine$. $\mathbf{W}$ and $\mathbf{b}$ are the weight matrix and bias term of the linear layer, respectively, and are shared throughout the network.

### 3.4 Cropping and extending signals

**Cropping**  is achieved at the partition level, meaning the precision of the cropping operation is directly influenced by the granularity of input space partitioning—smaller partitions enable more precise cropping. *Cropping involves identifying local sub-networks associated with the selected partitions for removal, and eliminating their weights.* In our experiments, we show that for image encoding, using partitions of $32 \times 32$ pixels results in respectable reconstruction accuracy, even surpassing the baseline INR. Recall that our objective for the cropping operation is to remove a number of weights proportionate to the removed segment of the signal, all without retraining. Since the weights of the global sub-network and the merge operator, collectively referred to as the

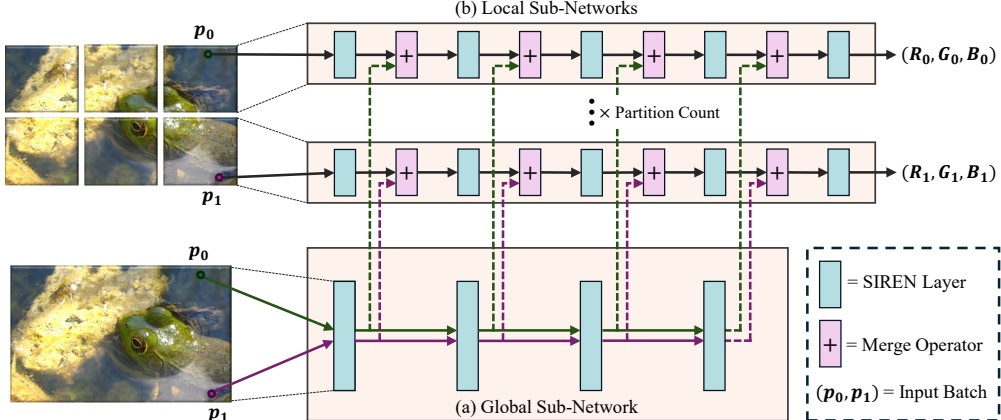

Figure 3: Illustration the Local-Global SIREN architecture and inference flow for two image coordinates $p_0, p_1$. The coordinates are passed through (a) the global sub-network and their partition's corresponding (b) local sub-network. Note that the coordinates are distinct elements in a batch, meaning that a coordinate's local features are only merged with the same coordinate's global features.

*global weights*, must be preserved, they should ideally encompass a relatively small part of the overall architecture. Additionally, opting for a larger global sub-network increases training and inference times due to the quadratic increase in the floating point operations associated with the fully connected layers. We found that allocating $5-15\%$ of the parameters to the global weights is sufficient for achieving the desired reconstruction accuracy, while also facilitating accelerated training and inference speeds, given by the dominance of the local sub-networks in the architecture.

**Extending** an encoded signal may also be achieved at the partition level. Since we cannot exclude a learning stage, the process is achieved by including additional local sub-networks, corresponding to the newly encoded partitions, followed by fine-tuning steps. As some partitions of the signal are already encoded, we use them to initialize the new weights. Similar to a mirror padding operation, at the signal's borders, we copy the local sub-network's weight symmetrically. We show in Section 4.4, that this process allows improved encoding of concatenated partitions compared to alternatives.

### 3.5 Additional properties

An interesting attribute of our proposed architecture is its capability to balance between the accuracy of the reconstructed signal and the efficiency, in terms of training and inference speeds, achieved by tuning the partition factors. **Increasing the number of partitions reduces neuron interconnectivity thus enhancing speed, while reducing them results in more accurate encoding.** This is explored in Section 4.5. Another interesting finding is that the local sub-networks can naturally exploit cases with significant differences between partitions. Consequently, our method manages to capture local details with fewer iterations. A qualitative example of this phenomenon is provided in Section 4.2.

### 3.6 Automatic partitioning and implementation

Our implementation offers either fixed or automatic signal partitioning configurations. With automatic partitioning, *the partition factors and sub-network hidden dimensions are automatically determined* based on the target partition size (e.g., $32 \times 32$ pixels in an image). This automatic approach reduces manual hyperparameter tuning and streamlines the training process of a Local-Global SIREN. Additional details are in Appendix A. We implement the local sub-networks using a custom locally connected (LC) layer, leveraging *torch*'s batched matrix-multiply operation–*bmm*, significantly outperforming a trivial implementation. During training, we use a batch containing the same number of sampled coordinates in each partition. If needed, one can use a portion of the LC layer's weights to reconstruct only the requested partitions.

**Limitations** While Torch-*bmm* has been useful in our case, it is known to be unstable, in terms of latency, across different GPUs. In addition, as cropping is possible at the partition level, it is limited in terms of flexibility. However, the extensive experiments presented highlight our method's potential.

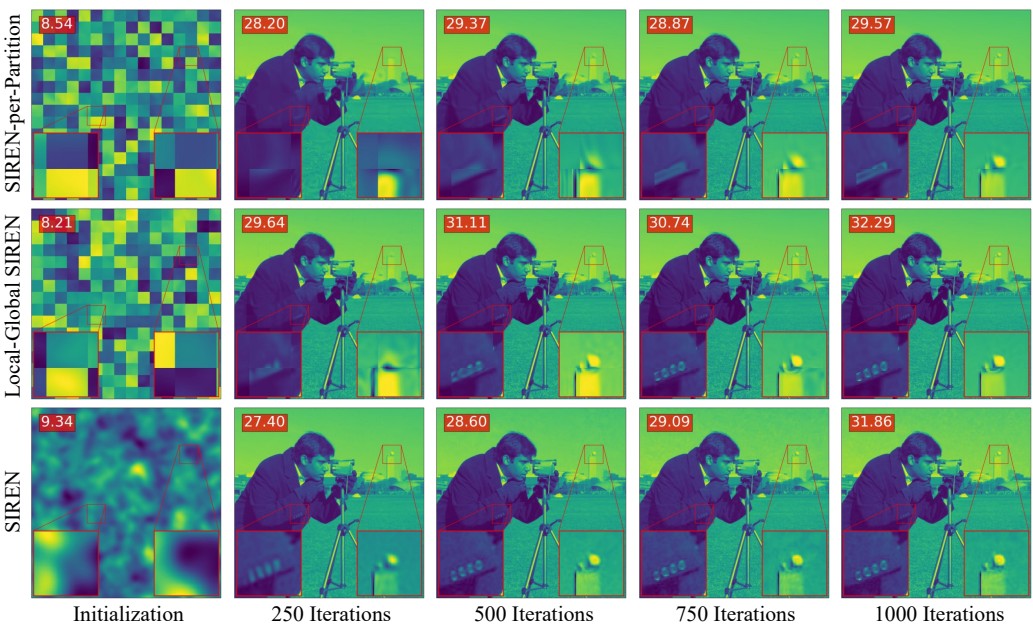

Figure 4: Encoded images throughout training iterations. PSNR values are at the top left of each image. Method names are on the left. Notice the artifacts in the SIREN-per-Partition method and the reduced noise in our approach compared to SIREN. For extended qualitative results of various signals and cropping operations, refer to `https://sites.google.com/view/local-global-inrs`.

## 4 Experiments

We evaluate our method on various signal encoding tasks and compare it to alternatives. To our knowledge, no other INR architecture supports cropping as we outlined earlier. Existing methods either necessitate an extra knowledge distillation step, or do not uniformly distribute weights across signal partitions, as in MINER [37]. Thus, we compare our approach to an *INR-per-Partition* and a full INR. We focus on SIREN as a baseline, and later examine INCODE for various tasks. We use roughly the same network size when comparing methods. **The full configuration and networks' size for all experiments is in Appendix B**. Since we manage to improve upon a full INR, comparing to a method based on knowledge distillation is redundant. For completeness, Appendix C shows why an approach like [35] does not trivially improve image encoding, and Appendix D demonstrates limitations of a hierarchical approach like MINER, in providing the discussed cropping capabilities.

We start with image, audio, video, and 3D shape encoding tasks. Local-Global SIRENs consistently outperform SIREN-per-Partition, and surpass SIREN while facilitating cropping in relatively fine granularity. Due to space constraints, 3D shape experiments are in Appendix E. Next, we present how Local-Global SIRENs allow extending a previously encoded signal, explore the tradeoff between latency and accuracy when choosing the partitioning factors, and explore the effects of global weight ratios. We further apply our Local-Global approach on INCODE, evaluating it on image encoding and various downstream tasks, showcasing its potential for boosting downstream performance. To ensure a fair comparison, we primarily followed the default omega values and learning rates from baseline methods. In some experiments, we made slight adjustments to the learning rate, ensuring that the selected values benefit all compared architectures. We ran the experiments multiple times on a single Nvidia RTX3090. For most experiments, we report the mean Structural Similarity (SSIM) with the mean and standard deviation of the Peak Signal-to-Noise Ratio (PSNR). Additional ablation experiments are in Appendix F.

### 4.1 Image encoding

We begin by evaluating our method on image encoding tasks. Qualitative results visualizing the training process on the famous $512 \times 512$ Cameraman image, using partition factors $C_0 = 16, C_1 = 16$ are shown in Figure 4. Next, from the DIV2K dataset [2], we have randomly selected a subset of

25 images which were downsampled by a factor of four before training. We have trained Local-Global SIRENs using both fixed and automatic partitioning configurations. For the fixed scheme, we set the partition factors to $C_0 = 16, C_1 = 16$, thus splitting images into partitions ranging from 21-32 pixels, with the global weights comprising $8\%$ of the network. The automatic scheme used a target partition size of $32 \times 32$ pixels, automatically selecting partition factors, and the global weights accounted for roughly $11\%$ of the network on average. Each network was trained for 2k iterations with a learning rate of $5 \cdot 10^{-4}$. Table 1 presents the mean SSIM and PSNR values on the DIV2K subset, demonstrating the efficacy Local-Global SIREN. These results indicate our approach significantly improves upon the SIREN-per-Partition baseline without requiring additional training steps.

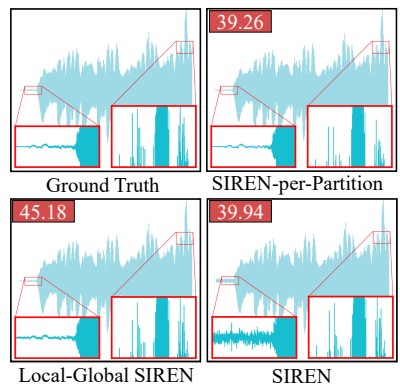

Table 1: Mean encoding results on 25 DIV2K images using five random seeds per image. Automatic partitioning uses partition factors $11 \le C_i \le 16$ to ensure $32 \times 32$ pixel partitions. SPP, LGS stand for SIREN-per-Parition and Local-Global SIREN, respectively.

| Method | Partition Factors | SSIM ↑ | PSNR (dB) ↑ |
|--------|------------------|--------|-------------|
| SPP | (16, 16) | 0.957 | 31.73 ± 0.63 |
| SPP | Auto | 0.955 | 31.90 ± 0.64 |
| LGS (ours) | (16, 16) | 0.968 | 33.94 ± 0.64 |
| LGS (ours) | Auto | **0.971** | **34.13 ± 0.59** |
| SIREN | - | 0.966 | 33.57 ± 0.65 |

Figure 5: Encoded *Bach* audio clips. Mean PSNR values using 10 random seeds are on the top left of each figure.

## 4.2 Audio encoding

We continue by evaluating our method on encoding audio clips, taken from [38]. We encode the first 7 seconds of Bach's Cello Suite No. 1: Prelude (*Bach*), and a 12 second clip of an actor counting 0-9 (*Counting*). We use a partitioning factor $C_0 = 32$, allowing cropping in roughly $220\,\mathrm{ms}$ and $370\,\mathrm{ms}$ intervals, respectively. The global weights account for $10.5\%$ of the network. We train each network for 1k iterations with a learning rate of $10^{-4}$. Local-Global SIRENs significantly outperform other methods, as seen in Figure 5. The local sub-networks can exploit the heterogeneous nature of audio partitions, showcasing less noise in silent regions, and a noticeable improvement in accuracy. Due to space constraints, quantitative results for both clips are in Appendix G.

## 4.3 Video encoding

Next, we evaluate our method on a video encoding task. We encode the 12 second cat video from [38], which has a spatial resolution of $512 \times 512$ and contains 300 frames. We use partition factors $C_0 = 5, C_1 = 8, C_2 = 8$, meaning we split the video in both the temporal and spatial dimensions. After training, the video can be cropped to $2.4\,\mathrm{s}$ intervals, of $64 \times 64$ pixels. The global weights take up $3.5\%$ of the network. We train each network for 5k iterations with a learning rate of $10^{-4}$. Another useful property of Local-Global SIRENs is its lower memory bandwidth constraints, allowing more pixels to be sampled per training iteration when encoding these large signals. We provide results for sampling $38 \times 10^{-4}\%$ of pixels, following the original paper, and demonstrate the effect of sampling more. The mean SSIM and PSNR values, computed on all frames, are pre-

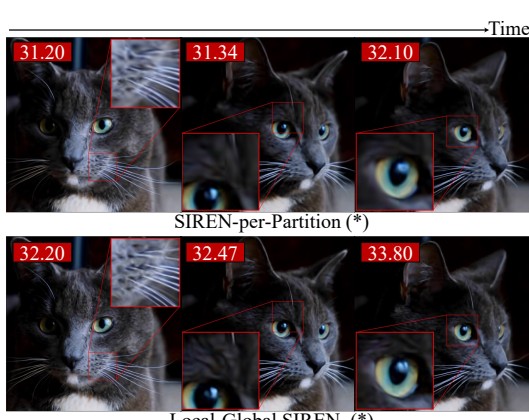

Figure 6: Three frames of an encoded video. PSNR is at the top left of each frame.

sented in Table 2. Qualitative results of three frames is in Figure 6. While SIREN-per-Partition achieves respectable results, artifacts remain apparent. Local-Global SIREN significantly reduces these artifacts and captures details better than SIREN. Cropped video examples are in Appendix H.

## 4.4 Extending an encoded image

We present how Local-Global SIRENs allow extending a previously encoded signal. We start by encoding the top $512 \times 256$ pixels of the Cameraman image from Section 4.1, using a SIREN and a Local-Global SIREN, both roughly 55% the size of the networks previously used. In our method, extending a signal is done by adding new local sub-networks, corresponding to the novel partitions, followed by fine-tuning. Thus, we add $16 \times 8$ local sub-networks. We compare this strategy with (1) fine-tuning the smaller SIREN on the entire signal, and (2) training a new full-sized SIREN. Figure 7 shows that, unsurprisingly, our approach outperforms both methods, since it can scale up the number of parameters while leveraging previous knowledge. Notice how fine-tuning the smaller SIREN stagnates fast, as it lacks capacity to encapsulate the entire signal. An extended plot is in Appendix I.

Table 2: Mean video encoding results, using 10 random seeds. (*) next to method stands for sampling $2 \cdot 10^{-2}\%$ of pixels in each iteration. SPP, LGS stand for SIREN-per-Parition and Local-Global SIREN, respectively.

| Method | SSIM ↑ | PSNR (dB) ↑ |
|--------|--------|-------------|
| SPP | 0.826 | 29.58 ± 0.02 |
| LGS (ours) | **0.854** | **30.28 ± 0.05** |
| SIREN | 0.815 | 29.71 ± 0.09 |
| SPP (*) | 0.854 | 30.83 ± 0.01 |
| LGS (*) (ours) | **0.888** | **31.91 ± 0.02** |

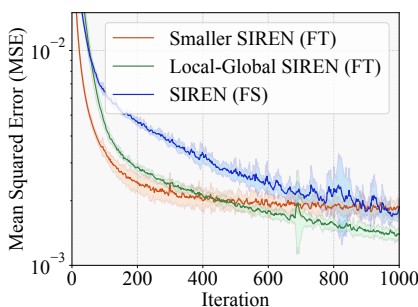

Figure 7: Log-scaled training MSE for extending a previously encoded image, using 10 seeds. Our approach (green) outperforms alternatives. FT, FS stand for fine-tuning and from-scratch, respectively.

## 4.5 Partitioning and global weights effects

There is an inherent tradeoff between latency and accuracy in Local-Global SIRENs, achieved by tuning the signal partition factors. To demonstrate the tradeoff, we encode the Cameraman image from Section 4.1 and the cat video from Section 4.3 using various partition factors. Results are in Table 3. Note how **enlarging the partition factors leads to faster training**, while decreasing the partition factors enhances reconstruction accuracy, surpassing SIREN. We additionally explore the extreme case of setting the partition factors to 1 in Appendix J. Next, we explore the effect of adjusting the ratio of global weights in the network. As shown in Table 4, larger global weights do not significantly boost accuracy and can sometimes have an adverse effect.

Table 3: Effect of partitioning on latency and accuracy. Results averaged on 10 seeds.

| Signal | Model | Partition Factors | MSE ↓ ($\cdot 10^{-4}$) | SSIM ↑ | PSNR (dB) ↑ | Train ↓ Time (s) |
|--------|-------|-------------------|-------------------------|--------|-------------|------------------|
| Image $512 \times 512$ | Local-Global SIREN (ours) | (2, 2) | **11.2** | **0.946** | **32.59 ± 0.52** | 74 |
| | | (4, 4) | 12.0 | **0.946** | 32.10 ± 0.47 | 40 |
| | | (8, 8) | 13.5 | 0.942 | 32.29 ± 0.42 | 26 |
| | | (16, 16) | 15.3 | 0.934 | 32.00 ± 0.39 | 22 |
| | | (32, 32) | 19.0 | 0.917 | 31.51 ± 0.28 | **15** |
| | SIREN | - | 18.4 | 0.914 | 31.17 ± 0.68 | 34 |
| Video $300 \times 512 \times 512$ | Local-Global SIREN (ours) | (5, 4, 4) | **32.8** | **0.862** | **30.95 ± 0.07** | 386 |
| | | (5, 8, 8) | 34.7 | 0.854 | 30.28 ± 0.05 | 284 |
| | | (5, 16, 16) | 41.8 | 0.834 | 29.52 ± 0.03 | **244** |
| | SIREN | - | 43.4 | 0.815 | 29.71 ± 0.09 | 2354 |

## 4.6 Local-Global INCODE

To further highlight the potential of the Local-Global approach, we examine the SOTA INCODE [20] INR as a potential baseline. INCODE contains a main MLP, harmonizer and a task-specific network. For a Local-Global INCODE, we modify the main MLP as seen earlier. The harmonizer, which configures properties of the nonlinearities, is adapted to output modulators for the local sub-networks, global sub-network and merge operator (i.e. 12 scalars). Since the task-specific model (e.g., ResNet [18]) is pre-trained and may remain frozen, we do not consider its weights as INR-specific. We compare Local-Global INCODE to INCODE and an INCODE-per-Partition on the image encoding task from Section 4.1, with results shown in Table 5. The global parameters constitute $15.5\%$ of the network, with the harmonizer taking an additional $3\%$. We train each network for 2k iterations, with a learning rate of $10^{-3}$ for local weights and $5.5 \cdot 10^{-4}$ for global weights.

Table 4: Encoding the Cameraman image (Section 4.1) and the cat video (Section 4.3) using different proportions of global weights, using 10 seeds.

| Signal | Global Weights (%) | SSIM ↑ | PSNR ↑ (dB) |
|---|---|---|---|
| Image | 11.6% | **0.934** | $32.00 \pm 0.39$ |
| | 22.9% | 0.931 | **$32.24 \pm 0.48$** |
| | 33.5% | 0.925 | $32.06 \pm 0.41$ |
| Video | 3.5% | **0.854** | **$30.28 \pm 0.05$** |
| | 10.2% | 0.850 | $30.04 \pm 0.07$ |

Table 5: Mean encoding results on 25 DIV2K images using five random seeds per image. We use partition factors $C_0 = 8, C_1 = 8$. IPP, LGI stand for INCODE-per-Partition and Local-Global INCODE, respectively.

| Method | SSIM ↑ | PSNR (dB) ↑ |
|---|---|---|
| IPP | 0.963 | $36.42 \pm 0.10$ |
| LGI (ours) | **0.974** | **$39.03 \pm 0.16$** |
| INCODE | 0.963 | $38.79 \pm 0.38$ |

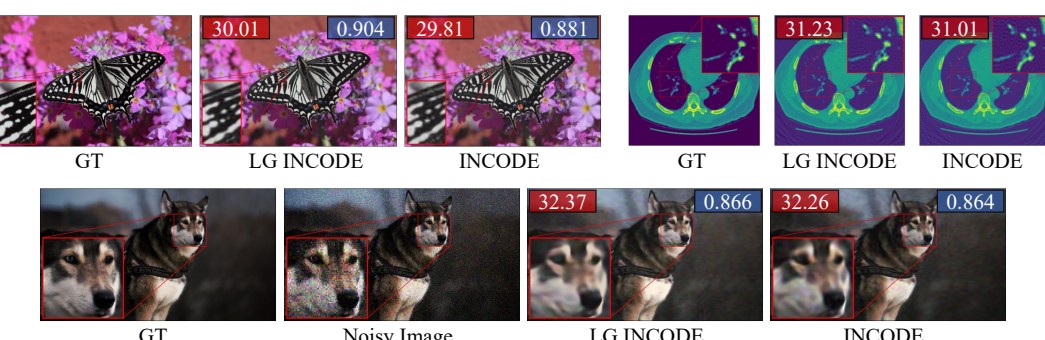

Figure 8: Local-Global (LG) INCODE applied to downstream tasks. Top-left: 4x image super-resolution, top-right: CT reconstruction, bottom: image denoising. Mean PSNR and SSIM values across 10 seeds are displayed in the top-left and top-right corners of each frame, respectively.

**Downstream tasks** INCODE showcased SOTA results on various downstream tasks. We demonstrate our approach's ability to boost downstream performance by replicating key experiments from the original paper: image denoising, super-resolution, and CT reconstruction. Qualitative results are in Figure 8, training configuration is in Appendix B.2 and full details and results are in Appendix K.

## 5 Conclusions and future work

In this paper we introduced Local-Global INRs, a novel architecture extension designed to seamlessly support cropping operations with a proportional weight decrease, without an additional pre-training or fine-tuning step. Local-Global INRs utilize both local and global contextual information, and have have surpassed alternative methods in terms of reconstruction accuracy. We further demonstrated how adjusting the signal partitioning allows for a balance between latency and the accuracy of the reconstructed signal. Furthermore, we have showcased instances where Local-Global INRs outperform the baseline INR's reconstruction accuracy, illustrated their capability to extend a previously encoded signal, and demonstrated their use in enhancing downstream performance. We believe our

proposed method represents a stride towards editable INRs and hope our findings will inspire further exploration into the design of INRs that inherently support modifications.

There are several directions for future work. Firstly, exploring additional architectural modifications, such as alternative merge operators, to refine our method's capabilities. Next, extending the Local-Global approach to INRs beyond MLP-based networks holds promise. Additionally, incorporating semantically meaningful partitions, as presented in [25], into our Local-Global architecture might further accelerate training and improve quality. Lastly, with Local-Global INRs representing each partition with a fixed amount of weights, there is potential in leveraging these networks for partition-based downstream tasks within INRs, as an extension to ideas in [13].

## Acknowledgements and disclosure of funding

The authors thank the Israeli Council for Higher Education (CHE) via the Data Science Research Center and the Lynn and William Frankel Center for Computer Science at BGU.

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

# A  Automatic partitioning

The automatic partitioning determines the hidden layer dimensions based on the target network size, global weights ratio, and the selected partition size. Pseudo-code is available below.

---

**Algorithm 1** Automatic Partitioning

---

 1: **function** AUTOMATICPARTITIONING(target_total_params, target_global_weight_ratio, target_partition_size, signal_resolution)
 2:      groups ← COMPUTENUMGROUPS(signal_resolution, target_partition_size)

 3:      global_hidden_dim ← FINDDIMENSION(target_total_params × target_global_weight_ratio)
 4:      global_weights ← COMPUTEGLOBALWEIGHTS(global_hidden_dim)

 5:      target_local_weights ← target_total_params - global_weights
 6:      local_hidden_dim ← FINDDIMENSION(target_local_weights, groups)
 7:      local_weights ← COMPUTELOCALWEIGHTS(local_hidden_dim, groups)

 8:      **while** global_weights + local_weights **not within** 1% of target_total_params **do**
 9:          **if** global_weights + local_weights > target_total_params **then**
10:              global_hidden_dim ← global_hidden_dim - 1
11:          **else**
12:              global_hidden_dim ← global_hidden_dim + 1
13:          **end if**
14:          global_weights ← COMPUTEGLOBALWEIGHTS(global_hidden_dim)
15:      **end while**

16:      **return** local_hidden_dim, global_hidden_dim
17: **end function**

18: **function** COMPUTENUMGROUPS(signal_resolution, target_partition_size)
     // Calculates the number of partitions
19: **end function**

20: **function** COMPUTEGLOBALWEIGHTS(global_hidden_dim)
     // Calculates global weights based on hidden dimension
21: **end function**

22: **function** COMPUTELOCALWEIGHTS(local_hidden_dim, groups)
     // Calculates local weights based on hidden dimension and groups
23: **end function**

24: **function** FINDDIMENSION(target_weight_count, groups)
     // Uses binary search to find the optimal dimension
25: **end function**

---

# B Complete experiment configuration

## B.1 Configuration for SIREN-based experiments

The configuration for all SIREN-based experiments in the paper is outlined in Table 6. For image encoding, audio encoding, and video encoding, we employ an AdamW scheduler [26] with learning rates of $5 \cdot 10^{-4}$, $10^{-4}$, and $10^{-4}$, respectively. We sample the entire signal in each iteration, when encoding images and audio. For videos, we sample a portion of pixels, as described in Section 4.3. The network hidden sizes remain fixed across layers, with all networks comprising five layers.

Table 6: Configuration of experiments based on SIREN. For the automatic partitioning, we report the mean number of parameters. LGS and SPP denote Local-Global SIREN (ours) and SIREN-per-Partition, respectively.

| Experiment Section | Method | Parameters | | | Hidden Sizes | | Partition Factors | | | # Iters |
|---|---|---|---|---|---|---|---|---|---|---|
| | | Total | Local | Global | Local | Global | $C_0$ | $C_1$ | $C_2$ | |
| Section 4.1 Cameraman (grayscale) | SIREN | 198.4k | - | 198.4k | - | 256 | - | - | - | 1k |
| | SPP | 200k | 200k | - | 15 | - | 16 | 16 | - | 1k |
| | LGS | 198.9k | 175.9k | 23k | 14 | 84 | 16 | 16 | - | 1k |
| Section 4.1 DIV2K (RGB) | SIREN | 198.9k | - | 198.9k | - | 256 | - | - | - | 2k |
| | SPP | 208.1k | 208.1k | - | 15 | - | 16 | 16 | - | 2k |
| | SPP(Auto) | 207.2k | 207.2k | - | 15-18 | - | 11-16 | 11-16 | - | 2k |
| | LGS | 199k | 183.6k | 15.4k | 14 | 68 | 16 | 16 | - | 2k |
| | LGS(Auto) | 200k | 180.4k | 19.7k | 14-17 | 69-86 | 11-16 | 11-16 | - | 2k |
| Section 4.2 Audio | SIREN | 198.1k | - | 198.1k | - | 256 | - | - | - | 1k |
| | SPP | 203k | 203k | - | 45 | - | 32 | - | - | 1k |
| | LGS | 198.2k | 177k | 20.7k | 42 | 72 | 32 | - | - | 1k |
| Section 4.3 Video | SIREN | 3.19M | - | 3.19M | - | 1030 | - | - | - | 5k |
| | SPP | 3.19M | 3.19M | - | 56 | - | 5 | 8 | 8 | 5k |
| | LGS | 3.19M | 3.08M | 111k | 55 | 180 | 5 | 8 | 8 | 5k |
| Section 4.4 (Half Image) | SIREN | 111k | - | 111k | - | 191 | - | - | - | 2k |
| | LGS | 111k | 88k | 23k | 14 | 84 | 16 | 8 | - | 2k |
| Section 4.4 (Full Image) | SIREN | 198.4k | - | 198.4k | - | 256 | - | - | - | 1k |
| | LGS | 198.9k | 175.9k | 23k | 14 | 84 | 16 | 16 | - | 1k |
| Section 4.5 Image Partitioning | LGS | 198.7k | 153.6k | 45k | 112 | 86 | 2 | 2 | - | 1k |
| | LGS | 197.5k | 173.7k | 23.8k | 59 | 72 | 4 | 4 | - | 1k |
| | LGS | 198.4k | 174.5k | 23.9k | 29 | 82 | 8 | 8 | - | 1k |
| | LGS | 198.9k | 175.9k | 23k | 14 | 84 | 16 | 16 | - | 1k |
| | LGS | 210.6k | 201.8k | 8.8k | 7 | 52 | 32 | 32 | - | 1k |
| Section 4.5 Video Partitioning | LGS | 3.18M | 3.05M | 131k | 111 | 180 | 5 | 4 | 4 | 5k |
| | LGS | 3.19M | 3.08M | 111k | 55 | 180 | 5 | 8 | 8 | 5k |
| | LGS | 3.25M | 3.15M | 104k | 27 | 180 | 5 | 16 | 16 | 5k |
| Section 4.5 Weights Image | LGS | 198.9k | 175.9k | 23k | 14 | 84 | 16 | 16 | - | 1k |
| | LGS | 199k | 153.3k | 45.7k | 13 | 120 | 16 | 16 | - | 1k |
| | LGS | 199k | 132.4k | 66.7k | 12 | 146 | 16 | 16 | - | 1k |
| Section 4.5 Weights Video | LGS | 3.19M | 3.08M | 111k | 55 | 180 | 5 | 8 | 8 | 5k |
| | LGS | 3.19M | 2.87M | 325.3k | 53 | 318 | 5 | 8 | 8 | 5k |
| Appendix E 3D Shape | SIREN | 198.7k | - | 198.7k | - | 256 | - | - | - | 27k |
| | SPP | 199.9k | 199.9k | - | 18 | - | 8 | 8 | 8 | 27k |
| | LGS | 199.2k | 179.7k | 19.5k | 17 | 76 | 8 | 8 | 8 | 27k |

## B.2 Configuration for INCODE-based experiments

The configuration for all INCODE-based experiments in the paper is outlined in Table 7. Following the original paper [20], we use the Adam optimizer [22] and a step-wise exponential learning rate scheduler across all experiments. For the DIV2K image encoding experiment, we use a learning rate of $10^{-3}$ for INCODE and the local weights of Local-Global INCODE. When training a Local-Global INCODE, we roughly halve the learning rate for the global weights. For the downstream tasks, we adopt the learning rates from the official implementation: $9 \cdot 10^{-4}$ for super-resolution, $1.5 \cdot 10^{-4}$ for image denoising, and $2 \cdot 10^{-4}$ for CT reconstruction. In each training iteration, the entire image is sampled. All networks comprise five layers, except for Local-Global INCODE in CT reconstruction, where we use three layers. The network hidden sizes remain fixed across layers.

Table 7: Configuration of experiments based on INCODE. The pre-trained ResNet parameters are excluded as they remain frozen during training. LGI and IPP denote Local-Global INCODE (ours) and INCODE-per-Partition, respectively.

| Experiment Section | Method | Parameters | | | Hidden Sizes | | Partition Factors | | | # Iters |
|---|---|---|---|---|---|---|---|---|---|---|
| | | Total | Local | Global | Local | Global | $C_0$ | $C_1$ | $C_2$ | |
| Section 4.6 DIV2K | INCODE | 205.3k | - | 205.3k | - | 256 | - | - | - | 2k |
| | IPP | 208.9k | 208.9k | - | 31 | - | 8 | 8 | - | 2k |
| | LGI | 205.2k | 166.8k | 38.4k | 28 | 96 | 8 | 8 | - | 2k |
| Section 4.6 Super Res. | INCODE | 205.3k | - | 205.3k | - | 256 | - | - | - | 2k |
| | LGI | 205.7k | 181.3k | 24.4k | 34 | 68 | 6 | 8 | - | 2k |
| Section 4.6 Denoising | INCODE | 201.9k | - | 201.9k | - | 256 | - | - | - | 1k |
| | LGI | 196.2k | 159.4k | 36.8k | 80 | 82 | 2 | 4 | - | 400 |
| Section 4.6 CT Recon. | INCODE | 327.5k | - | 327.5k | - | 326 | - | - | - | 2k |
| | LGI | 328.1k | 220k | 108.1k | 230 | 300 | 1 | 2 | - | 2k |

# C   Training with knowledge distillation

Training a compact INR for each partition in the signal is a straightforward approach for achieving the desired cropping and extension capabilities. However, as seen across all the experiments in the paper, the INR-per-Partition approach is subpar in terms of reconstruction accuracy and exhibits undesired artifacts in the reconstructed signal. Our approach solves this issue by learning both global and local contextual features, even surpassing the baseline INR accuracy. In methods based on NeRF [29], such as [35, 15], this is solved using knowledge distillation. Initially, a large NeRF is trained to encode the entire signal. Subsequently, the encoded representation is distilled into an ensemble of compact NeRFs using novel rendered views from the large NeRF. While this approach facilitates rapid rendering, it involves a more intricate training process, demanding additional time to train both the full NeRF and the compact NeRFs.

We aim to avoid this additional training step. However, for completeness, we explore a similar approach for encoding images using SIREN. We trained a full SIREN on the subset of 25 DIV2K images presented in Section 4.1. Next, for each trained SIREN, we decoded the image upsampled by $\times 2$ by sampling novel pixels between each two original pixels (e.g., for an image of size $512 \times 512$, we sample intermediate novel pixels, resulting in a $1023 \times 1023$ image). Subsequently, we trained a SIREN-per-Partition using the original pixels and the novel pixels extracted from the full SIREN. The results are presented in Table 8. Unfortunately, the reconstruction accuracy degrades in terms of both SSIM and PSNR when training using the novel pixels. We have tried various configurations, training iterations, checkpoint selection methods, loss weighting schemes, and general hyperparameter optimization; however, the results remained consistent and did not improve. *Note that even if the results improved due to knowledge distillation, they would likely be bounded by a full SIREN, which our Local-Global SIRENs manage to surpass.* We hypothesize that the methods in [35, 15] work due to the unique characteristic of NeRF, which is novel view rendering. Thus, these methods may not necessarily work on other domains and INR architectures.

Table 8: Results for encoding 25 images from DIV2K. Each image is encoded using five random seeds.

| Method | Partition Factors | Distillation | SSIM ↑ | PSNR ↑ (dB) |
|---|---|---|---|---|
| SIREN-per-Partition | (16, 16) | X | 0.955 | 31.90 ± 0.64 |
| SIREN-per-Partition | (16, 16) | ✓ | 0.953 | 31.64 ± 0.50 |
| Local-Global SIREN (ours) | (16, 16) | - | **0.971** | **34.13 ± 0.59** |
| SIREN | - | - | 0.966 | 33.57 ± 0.65 |

# D Comparison with MINER

MINER [37] employs multiscale coarse-to-fine INRs, encoding the Laplacian pyramid of signals. To enhance training speed, MINER dynamically selects regions needing finer details. While compelling, this approach results in an uneven weight distribution throughout the signal partitions, preventing cropping with relative weight reduction. To illustrate this, we have selected an image from the DIV2K dataset and encoded it with MINER and a Local-Global SIREN. The results are shown in Figure 9. Consider the scenario where one wishes to crop the image's boundary to center the person, as depicted in the figure. The border constitutes approximately 29% of the entire image. With MINER, most of the image border is encoded using coarse INRs, which cannot be removed. Consequently, only 2.8% of the entire weights can be discarded. However, using a Local-Global SIREN, one can discard the entire local weights corresponding to the border, reducing approximately 27% of the INR weights. This example highlights that while intriguing, MINER's coarse-to-fine approach cannot serve as a general solution for croppable INRs. Additionally, Local-Global SIRENs offer two technical advantages over MINER: first, MINER's implementation only supports signals with equal spatial dimensions (i.e., square-shaped images or cubic 3D scenes); second, since MINER dynamically allocates weights to different signal areas, the overall parameter count is challenging to calibrate and varies significantly between signals of the same size. Figure 10 illustrates this issue. This issue does not affect Local-Global SIRENs, as the weights are distributed equally, and the neural network structure is fixed.

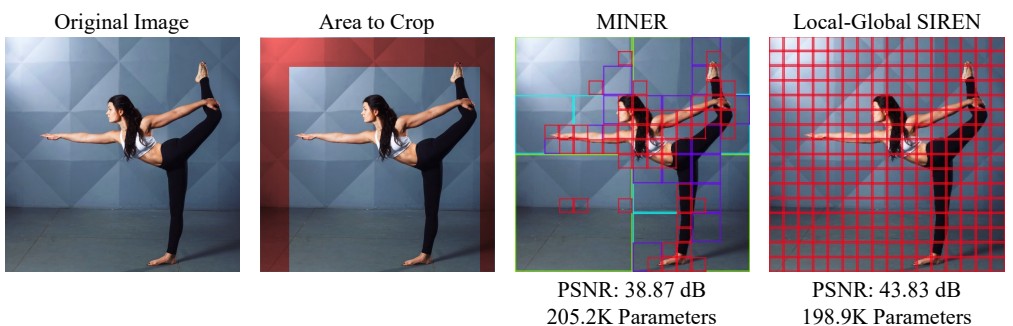

| Original Image | Area to Crop | MINER | Local-Global SIREN |
|---|---|---|---|
| | | PSNR: 38.87 dB | PSNR: 43.83 dB |
| | | 205.2K Parameters | 198.9K Parameters |

Figure 9: Illustration of cropping an image using MINER and Local-Global SIREN (ours). From left to right: (1) The original image, (2) The area to be cropped out of the encoded signal (roughly 29% of the image), (3) MINER reconstruction, with colors indicating the coarse-to-fine scale and weight distribution. Only 2.9% of MINER weights can be discarded, and (4) Local-Global SIREN reconstruction with uniform scale, allowing 27% of the INR weights to be discarded.

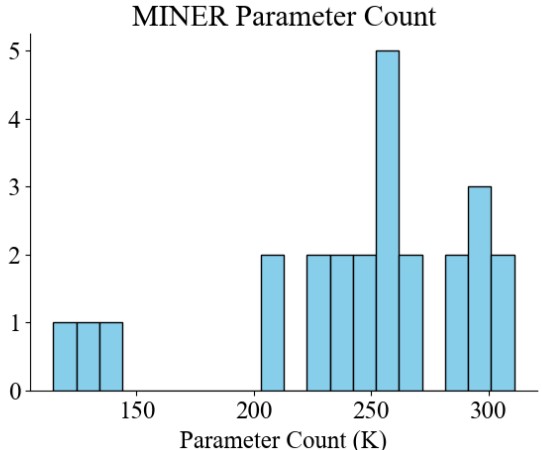

Figure 10: MINER parameter count histogram when encoding a subset of 25 images from the DIV2K dataset (as described in Section 4.1) with the exact same configuration.

# E  3D shape encoding

We evaluate our method on a 3D shape encoding task using the Lucy data from the Stanford 3D Scanning Repository [1]. We follow the data processing strategy presented in [36], and use the implementation from [20]. Points are sampled on a $512 \times 512 \times 512$ grid. Values of 1 are given to voxels within the object and 0 to voxels outside. We use partition factors $C_0 = 8, C_1 = 8, C_2 = 8$, meaning we split the grid into partitions of $64 \times 64 \times 64$ voxels. We assign local sub-networks only to non-empty partitions, and the global weights account for roughly 10% of the network. In each iteration we sample $10^5$ voxels, and train each network for roughly 27k iterations (20 epochs) using a learning rate of $10^{-4}$ and a step-wise exponential learning rate scheduler. We measure the Intersection over Union (IoU) between the encoded and original shape. Results are presented in Figure 11. Complete configuration is in Appendix B.1.

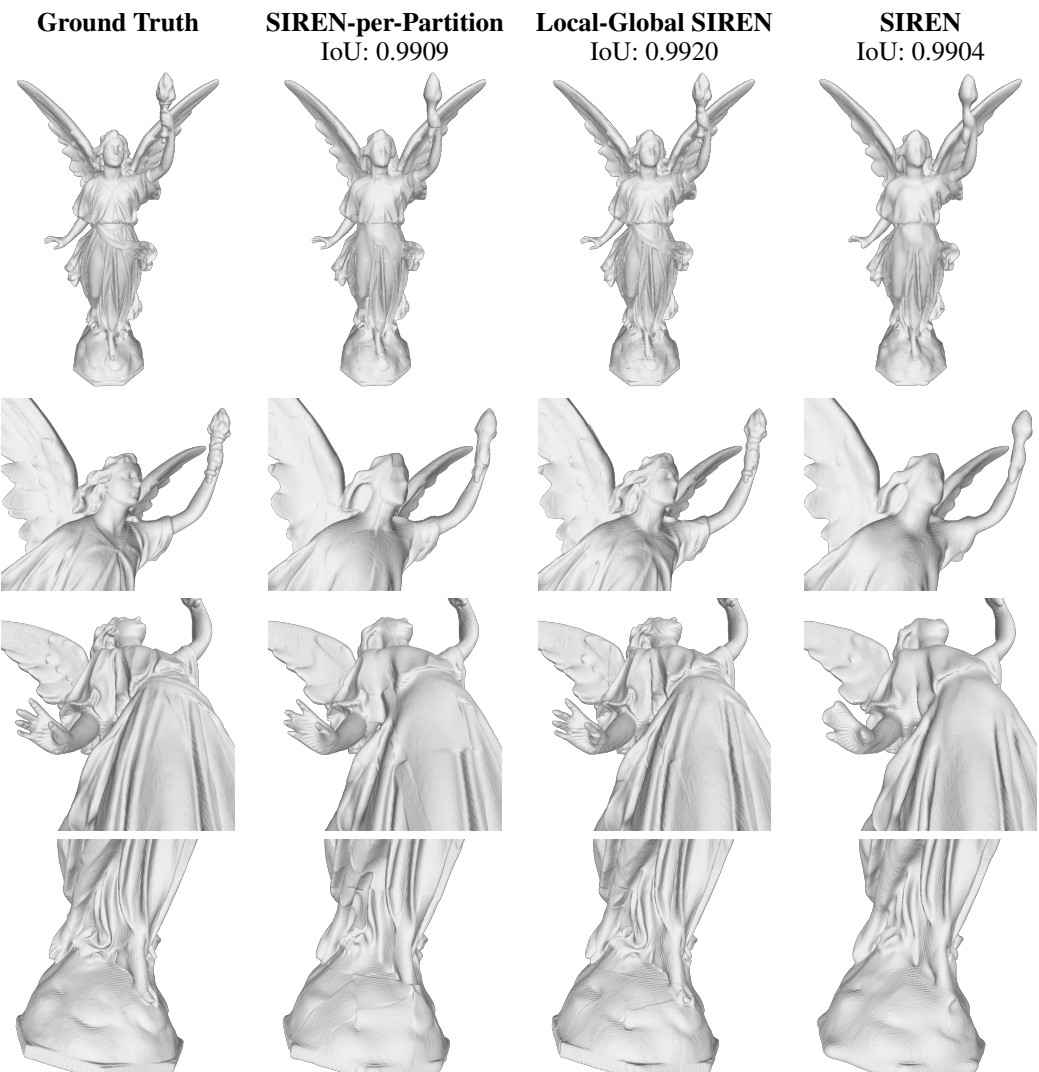

| **Ground Truth** | **SIREN-per-Partition**
IoU: 0.9909 | **Local-Global SIREN**
IoU: 0.9920 | **SIREN**
IoU: 0.9904 |

Figure 11: 3D shape encoding results. On the top of each column is the method name and achieved IoU score. Notice how Local-Global SIREN (ours) captures finer details with fewer artifacts than other methods, while facilitating cropping the shape to $64 \times 64 \times 64$ partitions.

---

[1] `https://graphics.stanford.edu/data/3Dscanrep/`

# F Additional ablation experiments

## F.1 Alternative merge operator

To highlight the importance of carefully crafting the merge operator, we explore an alternative, more efficient option. We utilize a smaller linear layer, that processes only global features. The new operator is given by:

$$\text{Merge}_{FC+Add}(\mathbf{L}, \mathbf{G}) = \mathbf{L} + \sigma(\mathbf{G} \cdot \mathbf{W} + \mathbf{b}). \tag{2}$$

Results in Table 9 indicate that, while more efficient, the alternative operator incurs a significant drop in accuracy.

Table 9: Comparison of merge operators when encoding the $512 \times 512$ Cameraman image. The original *Concat+FC* operator is compared against the more efficient *FC+Add*. Both use partition factors $C_0 = 16, C_1 = 16$, with the global sub-network hidden dimension set to 84 and the local sub-network dimension set to 14, resulting in a similar number of parameters. Averaged on 10 seeds.

| Merge Operator | SSIM ↑ | PSNR ↑ (dB) | Train Time (s) |
|---|---|---|---|
| *Concat+FC* | **0.934** | **32.00 ± 0.39** | 22 |
| *FC+Add* | 0.901 | 30.21 ± 0.48 | **20** |

## F.2 Network depth

Our chosen merge operator adds a learned layer. As a result, instead of $L$ layers, as in the original SIREN, our network can be seen as having $2L$ layers. To emphasize that the improved accuracy is a result of capturing both local and global contextual features, rather than depth, we train a SIREN-per-Partition with additional hidden layers. The results are presented in Table 10.

Table 10: Evaluating the effects of increasing the network depth for the SIREN-per-Partition method when encoding the $512 \times 512$ Cameraman image. We use partition factors $C_0 = 16, C_1 = 16$, and the results are averaged over 10 random seeds. The findings suggest that the effectiveness of Local-Global SIRENs is a result of merging local and global contextual features, rather than the number of nonlinearities in the architecture.

| Method | # Params | # Layers | SSIM ↑ | PSNR ↑ (dB) |
|---|---|---|---|---|
| SIREN-per-Partition | 194K | 10 | 0.902 | 31.24 ± 0.36 |
| SIREN-per-Partition | 236K | 10 | 0.916 | 31.67 ± 0.59 |
| Local-Global SIREN (ours) | 199K | 5 | **0.934** | **32.00 ± 0.39** |

# G    Audio quantitative results

Table 11: Audio encoding results after 1k training iterations. Averaged on 10 seeds.

| Audio Clip | Method | $C_0$ | MSE $(\cdot 10^{-5})\downarrow$ | PSNR (dB) $\uparrow$ |
|---|---|---|---|---|
| | SIREN-per-Partition | 32 | 12 | 39.26 ± 0.30 |
| **Bach** (7s) | Local-Global SIREN (ours) | 32 | **3** | **45.18 ± 0.99** |
| | SIREN | - | 10 | 39.94 ± 0.75 |
| | SIREN-per-Partition | 32 | 75 | 31.24 ± 0.19 |
| **Counting** (12s) | Local-Global SIREN (ours) | 32 | **48** | **33.18 ± 0.34** |
| | SIREN | - | 62 | 32.07 ± 0.32 |

Table 12: Audio encoding results after 5k training iterations. Averaged on 10 seeds.

| Audio Clip | Method | $C_0$ | MSE $(\cdot 10^{-5})\downarrow$ | PSNR (dB) $\uparrow$ |
|---|---|---|---|---|
| | SIREN-per-Partition | 32 | 2.4 | 46.20 ± 0.34 |
| **Bach** (7s) | Local-Global SIREN (ours) | 32 | **0.8** | **50.83 ± 0.55** |
| | SIREN | - | 1.3 | 48.93 ± 0.60 |
| | SIREN-per-Partition | 32 | 44 | 33.56 ± 0.08 |
| **Counting** (12s) | Local-Global SIREN (ours) | 32 | **28** | **35.56 ± 0.42** |
| | SIREN | - | 38 | 34.20 ± 0.13 |

# H   Video cropping examples

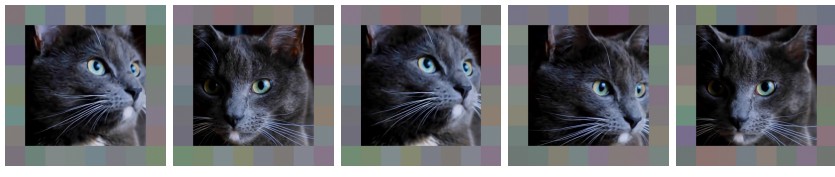

(a) Spatial cropping along the borders across all frames. Remaining parameters: 1.84M.

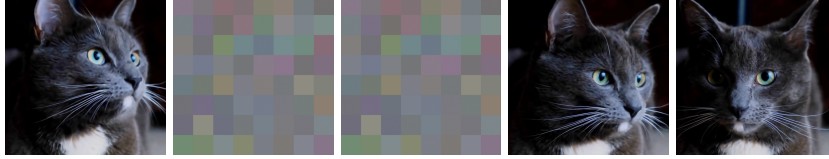

(b) Temporal cropping where frames of roughly 5(s) are removed. Remaining parameters: 1.96M.

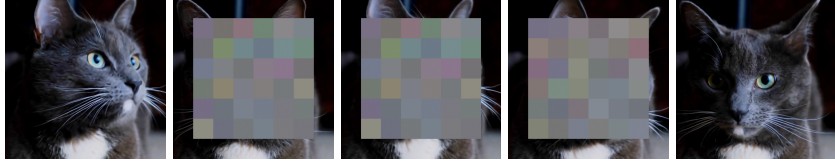

(c) Combined temporal and spatial cropping, where frames of roughly 7(s) are cropped in specific spatial areas. Remaining parameters: 2.5M.

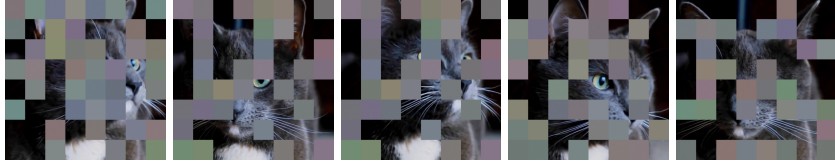

(d) Random cropping across different frames and spatial areas. Remaining parameters: 1.65M.

Figure 12: Cropping a video encoded in a Local-Global SIREN, containing 3.19M parameters, with different cropping strategies.

# I  Encoded image extension

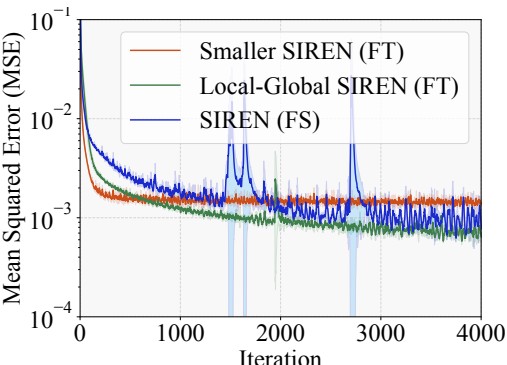

Figure 13: Log-scaled training MSE for extended a previously encoded image when significantly increasing training iterations, using 3 seeds. We pre-train and fine-tune for $4k$ iterations.

# J  Setting the partition factors to 1

While our Local-Global SIREN with partition factors of $\forall i, C_i = 1$ might appear similar to a regular SIREN, it is important to note that the two architectures are significantly different. In this extreme case, Local-Global SIREN still involves two networks with features intertwined throughout the forward pass. This intricate architecture affects performance, as seen in Table 13, for the Cameraman image encoding task discussed previously. Additionally, setting $\forall i, C_i = 1$ is somewhat analogous to INCODE, where a small modulator network augments the features of a large network. Thus, it is not surprising that Local-Global SIREN in this case achieves high reconstruction quality, even though it does not explicitly perform local feature learning.

Table 13: Effect of setting $C_0, C_1 = 1$ when encoding the Cameraman image. Averaged on 10 seeds.

| Model | Partition Factors | MSE $\downarrow$ $(\cdot 10^{-4})$ | SSIM $\uparrow$ | PSNR (dB) $\uparrow$ | Train $\downarrow$ Time (s) |
|---|---|---|---|---|---|
| Local-Global SIREN (ours) | (1, 1) | **12.9** | **0.939** | **32.52 ± 0.66** | 87 |
| SIREN | - | 18.4 | 0.914 | 31.17 ± 0.68 | 34 |

# K    Downstream tasks with Local-Global INCODE

For all downstream tasks, we replicate the setup from [20] using the official code implementation. Qualitative results are in Figure 8. Quanititative results are in Table 14. Full network configuration is in Appendix B.2. In addition, Table 15 presents results from an extended super-resolution experiment on 4x downsampled versions of the DIV2K subset, as outlined in Section 4.1.

**Image Denoising**    We use an image from the DIV2K dataset [2], resized to $512 \times 288 \times 3$ pixels. To create a noisy image simulating realistic sensor measurement, we apply readout and photon noise. Specifically, the mean photon count ($\tau$) is set to 40, and the readout noise is set to a signal-to-noise ratio of 2. Since the objective is to denoise the image using an INR, and we wish to focus on reconstruction accuracy, we choose coarse partition factors for Local-Global INCODE, specifically $C_0 = 4, C_1 = 2$.

**CT Reconstruction**    We use a publicly available $256 \times 256$ CT lung image from the Kaggle Lung Nodule Analysis dataset [2]. A sinogram is generated from the ground truth image using the radon transform with 180 projection measurements. The INRs aim to predict a reconstructed CT image from the sinogram data. To guide the model toward generating CT images with reduced artifacts, the loss function is computed between the sinograms of the INR's predicted output and the ground truth sinogram derived from the original image. Since the objective is to learn representations that align with the underlying projection data, we focus only on reconstruction accuracy, and use partition factors $C_0 = 2, C_1 = 1$ for Local-Global INCODE. This partitioning strategy vertically divides the image into two halves, with each partition containing one of the lungs. As seen in the experiments throughout the paper, coarser partitions enhance quality while finer ones improve cropping capabilities and latency.

**Image Super-Resolution**    We first replicate the experiment from [20], using an image from the DIV2K dataset [2] resized to $1344 \times 2016 \times 3$ pixels. We train INCODE and Local-Global INCODE to encode a 4x downsampled version of the image. During evaluation, we use the inherent interpolation capabilities of INRs to generate the full resolution image. We use partition factors $C_0 = 6, C_1 = 8$, splitting the image to partitions of $56 \times 63$ pixels. Finally, we extend this setup to a larger-scale experiment on the previously mentioned DIV2K subset, maintaining the same configuration.

Table 14: Mean results for downstream tasks across 10 random seeds, replicated from [20].

| Task | Method | SSIM ↑ | PSNR (dB) ↑ |
|------|--------|--------|-------------|
| Image Denoising | Local-Global INCODE | **0.866** | **32.37 ± 0.08** |
|  | INCODE | 0.864 | 32.26 ± 0.06 |
| CT Reconstruction | Local-Global INCODE | - | **31.23 ± 0.14** |
|  | INCODE | - | 31.01 ± 0.14 |
| Image Super-Resolution | Local-Global INCODE | **0.904** | **30.01 ± 0.11** |
|  | INCODE | 0.881 | 29.81 ± 0.11 |

Table 15: Extended mean super-resolution results on a subset of 25 DIV2K images.

| Task | Method | SSIM ↑ | PSNR (dB) ↑ |
|------|--------|--------|-------------|
| Image Super-Resolution | Local-Global INCODE | **0.734** | **26.76** |
|  | INCODE | 0.703 | 26.68 |

---

[2]https://luna16.grand-challenge.org/

# NeurIPS Paper Checklist

1. **Claims**

   Question: Do the main claims made in the abstract and introduction accurately reflect the paper's contributions and scope?

   Answer: [Yes]

   Justification: The abstract and introduction accurately reflect the key claims and contributions of the paper. Claim (1) that our proposed architecture can remove sections of the encoded signal with a proportional INR weight decrease is an inherent part of our proposed architecture, demonstrated in Figure 1 and explained in Section 3.4. Claim (2) of accelerated training is quantified in Table 3. Claim (3) that our approach can enhance encoding over the baseline INR is evidenced throughout the signal encoding experiments (e.g. Table 1). Claim (4) that our architecture supports extension of previously encoded signals is shown in Figure 7. Claim (5) of improved downstream performance is demonstrated in Figure 8 and Table 14. Claim (6) that our technique can be applied to modern INRs like INCODE is validated in Section 4.6.

   Guidelines:

   - The answer NA means that the abstract and introduction do not include the claims made in the paper.
   - The abstract and/or introduction should clearly state the claims made, including the contributions made in the paper and important assumptions and limitations. A No or NA answer to this question will not be perceived well by the reviewers.
   - The claims made should match theoretical and experimental results, and reflect how much the results can be expected to generalize to other settings.
   - It is fine to include aspirational goals as motivation as long as it is clear that these goals are not attained by the paper.

2. **Limitations**

   Question: Does the paper discuss the limitations of the work performed by the authors?

   Answer: [Yes]

   Justification: We discuss the limitations of our proposed method, covering both implementation details and design aspects, in a dedicated highlighted paragraph at the end of Section 3.6. To demonstrate the robustness of our results, we report the number of random seeds used for each experiment. Additionally, we validate our approach across various domains and instances, providing confidence in of our conclusions. Furthermore, we investigate the trade-off between the number of partitions and the accuracy of the reconstructed signal, showing that while significantly increasing number of partitions reduces training time, it also degrades the quality of the reconstructed signal (see Table 3). Regarding privacy and fairness considerations, we believe these issues are not specific to our area of research, but rather general concerns applicable to the machine learning field as a whole.

   Guidelines:

   - The answer NA means that the paper has no limitation while the answer No means that the paper has limitations, but those are not discussed in the paper.
   - The authors are encouraged to create a separate "Limitations" section in their paper.
   - The paper should point out any strong assumptions and how robust the results are to violations of these assumptions (e.g., independence assumptions, noiseless settings, model well-specification, asymptotic approximations only holding locally). The authors should reflect on how these assumptions might be violated in practice and what the implications would be.
   - The authors should reflect on the scope of the claims made, e.g., if the approach was only tested on a few datasets or with a few runs. In general, empirical results often depend on implicit assumptions, which should be articulated.
   - The authors should reflect on the factors that influence the performance of the approach. For example, a facial recognition algorithm may perform poorly when image resolution is low or images are taken in low lighting. Or a speech-to-text system might not be

used reliably to provide closed captions for online lectures because it fails to handle technical jargon.

- The authors should discuss the computational efficiency of the proposed algorithms and how they scale with dataset size.
- If applicable, the authors should discuss possible limitations of their approach to address problems of privacy and fairness.
- While the authors might fear that complete honesty about limitations might be used by reviewers as grounds for rejection, a worse outcome might be that reviewers discover limitations that aren't acknowledged in the paper. The authors should use their best judgment and recognize that individual actions in favor of transparency play an important role in developing norms that preserve the integrity of the community. Reviewers will be specifically instructed to not penalize honesty concerning limitations.

3. **Theory Assumptions and Proofs**

Question: For each theoretical result, does the paper provide the full set of assumptions and a complete (and correct) proof?

Answer: [NA]

Justification: We do not cover theoretical results in our paper, but rather justify our design choices through various experimental scenarios.

Guidelines:

- The answer NA means that the paper does not include theoretical results.
- All the theorems, formulas, and proofs in the paper should be numbered and cross-referenced.
- All assumptions should be clearly stated or referenced in the statement of any theorems.
- The proofs can either appear in the main paper or the supplemental material, but if they appear in the supplemental material, the authors are encouraged to provide a short proof sketch to provide intuition.
- Inversely, any informal proof provided in the core of the paper should be complemented by formal proofs provided in appendix or supplemental material.
- Theorems and Lemmas that the proof relies upon should be properly referenced.

4. **Experimental Result Reproducibility**

Question: Does the paper fully disclose all the information needed to reproduce the main experimental results of the paper to the extent that it affects the main claims and/or conclusions of the paper (regardless of whether the code and data are provided or not)?

Answer: [Yes]

Justification: We provide details on the full training setup used for all experiments in Appendix B. These details include the optimizer, learning rates, network architecture, number of training iterations, and other hyperparameter settings. Additionally, we have released the code implementation of our proposed architecture, the training loop, and evaluation code, along with a detailed README file to allow others to run and reproduce our results. The specific signals (images, audio, video, etc.) that were used for training our models are also included in the supplementary materials.

Guidelines:

- The answer NA means that the paper does not include experiments.
- If the paper includes experiments, a No answer to this question will not be perceived well by the reviewers: Making the paper reproducible is important, regardless of whether the code and data are provided or not.
- If the contribution is a dataset and/or model, the authors should describe the steps taken to make their results reproducible or verifiable.
- Depending on the contribution, reproducibility can be accomplished in various ways. For example, if the contribution is a novel architecture, describing the architecture fully might suffice, or if the contribution is a specific model and empirical evaluation, it may be necessary to either make it possible for others to replicate the model with the same dataset, or provide access to the model. In general. releasing code and data is often

one good way to accomplish this, but reproducibility can also be provided via detailed instructions for how to replicate the results, access to a hosted model (e.g., in the case of a large language model), releasing of a model checkpoint, or other means that are appropriate to the research performed.

- While NeurIPS does not require releasing code, the conference does require all submissions to provide some reasonable avenue for reproducibility, which may depend on the nature of the contribution. For example
    (a) If the contribution is primarily a new algorithm, the paper should make it clear how to reproduce that algorithm.
    (b) If the contribution is primarily a new model architecture, the paper should describe the architecture clearly and fully.
    (c) If the contribution is a new model (e.g., a large language model), then there should either be a way to access this model for reproducing the results or a way to reproduce the model (e.g., with an open-source dataset or instructions for how to construct the dataset).
    (d) We recognize that reproducibility may be tricky in some cases, in which case authors are welcome to describe the particular way they provide for reproducibility. In the case of closed-source models, it may be that access to the model is limited in some way (e.g., to registered users), but it should be possible for other researchers to have some path to reproducing or verifying the results.

5. **Open access to data and code**

    Question: Does the paper provide open access to the data and code, with sufficient instructions to faithfully reproduce the main experimental results, as described in supplemental material?

    Answer: [Yes]

    Justification: We have released the code implementation of our proposed architecture, the training loop, and evaluation code, along with a detailed README file to allow others to run and reproduce our results. The specific signals (images, audio, video, etc.) that were used for training our models are also included in the supplementary materials.

    Guidelines:

    - The answer NA means that paper does not include experiments requiring code.
    - Please see the NeurIPS code and data submission guidelines (`https://nips.cc/public/guides/CodeSubmissionPolicy`) for more details.
    - While we encourage the release of code and data, we understand that this might not be possible, so "No" is an acceptable answer. Papers cannot be rejected simply for not including code, unless this is central to the contribution (e.g., for a new open-source benchmark).
    - The instructions should contain the exact command and environment needed to run to reproduce the results. See the NeurIPS code and data submission guidelines (`https://nips.cc/public/guides/CodeSubmissionPolicy`) for more details.
    - The authors should provide instructions on data access and preparation, including how to access the raw data, preprocessed data, intermediate data, and generated data, etc.
    - The authors should provide scripts to reproduce all experimental results for the new proposed method and baselines. If only a subset of experiments are reproducible, they should state which ones are omitted from the script and why.
    - At submission time, to preserve anonymity, the authors should release anonymized versions (if applicable).
    - Providing as much information as possible in supplemental material (appended to the paper) is recommended, but including URLs to data and code is permitted.

6. **Experimental Setting/Details**

    Question: Does the paper specify all the training and test details (e.g., data splits, hyperparameters, how they were chosen, type of optimizer, etc.) necessary to understand the results?

    Answer: [Yes]

Justification: For each experiment, we provide the important details necessary to understand and appreciate the results. For instance, Section 4.1 outlines the partition factors, global weights ratios, partition size in pixels, number of images used, their sources, number of random seeds, learning rate, and number of training iterations. We follow a similar approach of including relevant experimental details across other sections as well. The complete training setup configurations and network hyperparameters used in all experiments are in Appendix B.

Guidelines:

- The answer NA means that the paper does not include experiments.
- The experimental setting should be presented in the core of the paper to a level of detail that is necessary to appreciate the results and make sense of them.
- The full details can be provided either with the code, in appendix, or as supplemental material.

7. **Experiment Statistical Significance**

Question: Does the paper report error bars suitably and correctly defined or other appropriate information about the statistical significance of the experiments?

Answer: [Yes]

Justification: We test our method on various types of signals, with each experiment being run multiple times. For images, we also experiment of a subset of 25 images from DIV2K, where we encode each of the images multiple times. We use PSNR as an encoding quality metric throughout all experiments, and add the standard deviation as well as its mean (see Section 4 for details, and all other tables for information on number of seeds and PSNR standard deviation, e.g, Table 5, Table 12). Due to space constraints, we did not add the standard deviation for SSIM.

Guidelines:

- The answer NA means that the paper does not include experiments.
- The authors should answer "Yes" if the results are accompanied by error bars, confidence intervals, or statistical significance tests, at least for the experiments that support the main claims of the paper.
- The factors of variability that the error bars are capturing should be clearly stated (for example, train/test split, initialization, random drawing of some parameter, or overall run with given experimental conditions).
- The method for calculating the error bars should be explained (closed form formula, call to a library function, bootstrap, etc.)
- The assumptions made should be given (e.g., Normally distributed errors).
- It should be clear whether the error bar is the standard deviation or the standard error of the mean.
- It is OK to report 1-sigma error bars, but one should state it. The authors should preferably report a 2-sigma error bar than state that they have a 96% CI, if the hypothesis of Normality of errors is not verified.
- For asymmetric distributions, the authors should be careful not to show in tables or figures symmetric error bars that would yield results that are out of range (e.g. negative error rates).
- If error bars are reported in tables or plots, The authors should explain in the text how they were calculated and reference the corresponding figures or tables in the text.

8. **Experiments Compute Resources**

Question: For each experiment, does the paper provide sufficient information on the computer resources (type of compute workers, memory, time of execution) needed to reproduce the experiments?

Answer: [Yes]

Justification: We specify at the beginning of Section 4 that all experiments were run on a single Nvidia RTX3090 GPU with 24GB of RAM. This indicates that replicating all our experiments requires no more than 24GB of available RAM. In Section 4.5, we report the

training time for image and video encoding tasks. While runtimes for other tasks are omitted due to space constraints, the video encoding task serves as an upper bound, being the largest signal encoded in the main paper.

- The answer NA means that the paper does not include experiments.
- The paper should indicate the type of compute workers CPU or GPU, internal cluster, or cloud provider, including relevant memory and storage.
- The paper should provide the amount of compute required for each of the individual experimental runs as well as estimate the total compute.
- The paper should disclose whether the full research project required more compute than the experiments reported in the paper (e.g., preliminary or failed experiments that didn't make it into the paper).

9. **Code Of Ethics**

Question: Does the research conducted in the paper conform, in every respect, with the NeurIPS Code of Ethics `https://neurips.cc/public/EthicsGuidelines`?

Answer: [Yes]

Justification: We have read through the code of ethics and confirm to it.

Guidelines:

- The answer NA means that the authors have not reviewed the NeurIPS Code of Ethics.
- If the authors answer No, they should explain the special circumstances that require a deviation from the Code of Ethics.
- The authors should make sure to preserve anonymity (e.g., if there is a special consideration due to laws or regulations in their jurisdiction).

10. **Broader Impacts**

Question: Does the paper discuss both potential positive societal impacts and negative societal impacts of the work performed?

Answer: [NA]

Justification: Our work focuses on advancing the technical capabilities of implicit neural representations for signal encoding and editing, which does not directly introduce novel societal impacts. However, we acknowledge that the responsible development and deployment of machine learning technologies requires consideration of potential risks and ethical concerns, which is relevant to the field as a whole.

Guidelines:

- The answer NA means that there is no societal impact of the work performed.
- If the authors answer NA or No, they should explain why their work has no societal impact or why the paper does not address societal impact.
- Examples of negative societal impacts include potential malicious or unintended uses (e.g., disinformation, generating fake profiles, surveillance), fairness considerations (e.g., deployment of technologies that could make decisions that unfairly impact specific groups), privacy considerations, and security considerations.
- The conference expects that many papers will be foundational research and not tied to particular applications, let alone deployments. However, if there is a direct path to any negative applications, the authors should point it out. For example, it is legitimate to point out that an improvement in the quality of generative models could be used to generate deepfakes for disinformation. On the other hand, it is not needed to point out that a generic algorithm for optimizing neural networks could enable people to train models that generate Deepfakes faster.
- The authors should consider possible harms that could arise when the technology is being used as intended and functioning correctly, harms that could arise when the technology is being used as intended but gives incorrect results, and harms following from (intentional or unintentional) misuse of the technology.
- If there are negative societal impacts, the authors could also discuss possible mitigation strategies (e.g., gated release of models, providing defenses in addition to attacks, mechanisms for monitoring misuse, mechanisms to monitor how a system learns from feedback over time, improving the efficiency and accessibility of ML).

11. **Safeguards**

    Question: Does the paper describe safeguards that have been put in place for responsible release of data or models that have a high risk for misuse (e.g., pretrained language models, image generators, or scraped datasets)?

    Answer: [NA]

    Justification: Our work does not release any pretrained models, generators, or scraped datasets that could pose risks of misuse. We utilize standard datasets for experiments, and cite the sources when introducing them in the paper. Therefore, we feel as if safeguards for responsible release are not applicable.

    Guidelines:

    - The answer NA means that the paper poses no such risks.
    - Released models that have a high risk for misuse or dual-use should be released with necessary safeguards to allow for controlled use of the model, for example by requiring that users adhere to usage guidelines or restrictions to access the model or implementing safety filters.
    - Datasets that have been scraped from the Internet could pose safety risks. The authors should describe how they avoided releasing unsafe images.
    - We recognize that providing effective safeguards is challenging, and many papers do not require this, but we encourage authors to take this into account and make a best faith effort.

12. **Licenses for existing assets**

    Question: Are the creators or original owners of assets (e.g., code, data, models), used in the paper, properly credited and are the license and terms of use explicitly mentioned and properly respected?

    Answer: [Yes]

    Justification: In our paper, we credit the creators and original owners of any existing assets (code, data, models) that were utilized. When introducing the datasets used for experiments, such as those detailed in Section 4.1 and Appendix E, we cite the relevant sources they were taken from. Similarly, for any existing codebases or specific data samples used as a foundation for our work, like the data samples mentioned in Section 4.3 and Section 4.2, we cite the source and clearly state the licenses (e.g., MIT) under which they were originally released in our supplementary material. Additionally, our released code package includes a README file that explicitly lists the MIT license and provides credits for any third-party codebases incorporated into our implementation.

    Guidelines:

    - The answer NA means that the paper does not use existing assets.
    - The authors should cite the original paper that produced the code package or dataset.
    - The authors should state which version of the asset is used and, if possible, include a URL.
    - The name of the license (e.g., CC-BY 4.0) should be included for each asset.
    - For scraped data from a particular source (e.g., website), the copyright and terms of service of that source should be provided.
    - If assets are released, the license, copyright information, and terms of use in the package should be provided. For popular datasets, `paperswithcode.com/datasets` has curated licenses for some datasets. Their licensing guide can help determine the license of a dataset.
    - For existing datasets that are re-packaged, both the original license and the license of the derived asset (if it has changed) should be provided.
    - If this information is not available online, the authors are encouraged to reach out to the asset's creators.

13. **New Assets**

    Question: Are new assets introduced in the paper well documented and is the documentation provided alongside the assets?

Answer: [Yes]

Justification: The relevant asset in our case is the code implementation, which can be used to recreate the experiments presented in the paper. The supplementary material, which includes the code, provides detailed documentation on the license and instructions on running the code.

Guidelines:

- The answer NA means that the paper does not release new assets.
- Researchers should communicate the details of the dataset/code/model as part of their submissions via structured templates. This includes details about training, license, limitations, etc.
- The paper should discuss whether and how consent was obtained from people whose asset is used.
- At submission time, remember to anonymize your assets (if applicable). You can either create an anonymized URL or include an anonymized zip file.

14. **Crowdsourcing and Research with Human Subjects**

Question: For crowdsourcing experiments and research with human subjects, does the paper include the full text of instructions given to participants and screenshots, if applicable, as well as details about compensation (if any)?

Answer: [NA]

Justification: This paper does not involve crowdsourcing experiments or research with human subjects.

Guidelines:

- The answer NA means that the paper does not involve crowdsourcing nor research with human subjects.
- Including this information in the supplemental material is fine, but if the main contribution of the paper involves human subjects, then as much detail as possible should be included in the main paper.
- According to the NeurIPS Code of Ethics, workers involved in data collection, curation, or other labor should be paid at least the minimum wage in the country of the data collector.

15. **Institutional Review Board (IRB) Approvals or Equivalent for Research with Human Subjects**

Question: Does the paper describe potential risks incurred by study participants, whether such risks were disclosed to the subjects, and whether Institutional Review Board (IRB) approvals (or an equivalent approval/review based on the requirements of your country or institution) were obtained?

Answer: [NA]

Justification:This paper does not involve crowdsourcing experiments or research with human subjects.

Guidelines:

- The answer NA means that the paper does not involve crowdsourcing nor research with human subjects.
- Depending on the country in which research is conducted, IRB approval (or equivalent) may be required for any human subjects research. If you obtained IRB approval, you should clearly state this in the paper.
- We recognize that the procedures for this may vary significantly between institutions and locations, and we expect authors to adhere to the NeurIPS Code of Ethics and the guidelines for their institution.
- For initial submissions, do not include any information that would break anonymity (if applicable), such as the institution conducting the review.

