# OpenReview forum: "Towards Croppable Implicit Neural Representations"
_NeurIPS.cc/2024/Conference — NeurIPS 2024 poster_

### Official Review · Reviewer_XqfY · 2024-06-28

**Soundness:** 2
**Presentation:** 3
**Contribution:** 2
**Rating:** 6
**Confidence:** 4

**Summary:**

This paper proposes Local-Global SIRENs, which partition the space into different regions and fit each region with smaller local INRs, leading to croppable INR by cropping the weights relative to the local regions. The model further proposes to use local and global feature extraction to improve the fitting performance. The experiments show that their method supports cropping INRs for Image, Audio, Video and CT.

**Strengths:**

1. The idea of using partition-based INRs for croppable INRs is novel. Croppable INRs are an interesting application for partition-based INRs.
2. The paper is well-written and easy to follow.
3. The cropping performance looks fancy and the fitting performance is slightly improved compared to SIREN due to the global feature extraction.

**Weaknesses:**

1. Cropping INRs is a natural property of partition-based INRs, which makes the contribution minor.
2. While the authors mention their method supports automatic partitioning, they do not show the detailed implementation of how to automatically partition the space.
3. Even though the authors have conducted an ablation study on partition factors, the range of experimented partition factors is not enough. I recommend the authors to try Partition
Factors as low as (2,2) and as high as (512, 512) for the 512 * 512 images and show how the partition affects the fitting performance.
4. Cropping based on simple grids may be impractical for real scenarios. I recommend the authors improve their method by implementing semantic segmentation-based cropping. See partition-based INRs with semantic segmentation [1].
5. I am confused about the conclusion that enlarging the partition factors leads to faster training while decreasing the partition factors enhances reconstruction accuracy (line 293). Since SIREN is just your Local-Global SIREN with partition factors (1,1), if your conclusion is right, SIREN should have better performance than your LG SIREN. As pointed out in [1], partition should generally improve the fitting performance with larger partition factors. And please provide some explanation about why increasing the number of partitions enhances overall speed.

[1] Liu, Ke, et al. "Partition speeds up learning implicit neural representations based on exponential-increase hypothesis." Proceedings of the IEEE/CVF International Conference on Computer Vision. 2023.

**Questions:**

1. Could you please discuss how to automatically partition the region with your method?
2. Could you please discuss whether your method can be extended to semantic segmentation-based cropping?

**Limitations:**

The limitations and potential negative societal impact of their work have been discussed.

---

> ### Author Rebuttal · Authors · 2024-08-05
>
> Dear reviewer, thank you for the constructive feedback and great reference. We appreciate the time and effort. Please see our comments and additional results below.
>
> **[W1] Contribution**
>
> While training an INR-per-partition does allow for cropping with a proportionate weight decrease, our extensive experiments across various modalities show that this method significantly degrades encoding quality and introduces unwanted artifacts. These artifacts are illustrated in Figures 4, 6, and 11, with quantitative results presented in Tables 1, 2, and 12. Our method not only addresses these issues but also **improves upon the baseline INR itself**. Local-Global INRs enable cropping capabilities while enhancing encoding quality and performance on downstream tasks.
>
> Although other INR-per-partition methods using larger or semantically-uniform partitions might improve over uniform partitions, they have two downsides: (1) They limit the flexibility of cropping. (2) Since the partitions are not fixed across various signals, future applications of partition-based downstream tasks become more difficult to achieve.
>
> **[W2/Q1] Automatic Partitioning**
>
> The automatic partitioning process is described briefly in Section 3.6 and involves a straightforward approach. Given a target partition size (e.g., 32x32 image pixels) and a required global weight ratio (typically around 10%), we compute the partition factors through simple division and determine the hidden dimensions of sub-networks using binary search. The implementation is provided in the `compute_partitions.py` module. We will include the relevant pseudocode in the final revision of the paper (please see pseudocode in the general top comment).
>
> **[W3] Partition Factors Range**
>
> We agree that extending the range of partition factors is valuable, though space constraints limited our initial revision. We have now included image encoding results for partition factors (2, 2), (32, 32), and (64, 64). The results are as follows:
> | Partition Factors | MSE (*10^-4) | SSIM | PSNR | Train Time (s) |
> |-------|----------|--------------|-------|---------|
> |(2,2)|**11.2**|**0.946**|**32.59 ± 0.52**|74|
> |(4,4)|12.0|**0.946**|32.10 ± 0.47|40|
> |(8,8)|13.5|0.942|32.29 ± 0.42|26|
> |(16,16)|15.3|0.934|32.00 ± 0.39|22|
> |(32,32)|19.0|0.917|31.51 ± 0.28|**15**|
> |(64,64)|31.9|0.868|29.56 ± 0.45|**15**|
>
> Using larger partition factors, as the reviewer mentioned, is not feasible as there are not enough weights in the network to distribute among the partitions. Further increasing partition sizes would require expanding the network capacity beyond the typical 200k parameters used in previous literature for image encoding tasks. We will include these extended experiments in the final revision of the paper.
>
> **[W4/Q2] Segmentation-based Cropping**
>
> Thank you for the suggestion. Implementing semantic segmentation-based cropping could indeed enhance the quality of our method by aligning partitions with meaningful image features. However, we believe integrating this approach into our current work is beyond the scope of this paper. We recognize that while semantic segmentation-based methods can improve the encoding quality and offer context-aware cropping, they may also limit cropping flexibility since only entire segments may be cropped (we briefly mentioned this in the first subsection of this rebuttal). We will definitely consider exploring this direction as part of our future research. Additionally, the paper you cited is highly relevant to our approach, and we will include it in both the related work and future work sections of our paper.
>
> **[W5] Quality-Latency Trade Off**
>
> Thank you for highlighting this point. It is important to clarify that while SIREN with partition factors of (1,1) might appear similar to our Local-Global SIREN, the two architectures are **significantly different**. LG-SIREN with (1,1) still involves two networks with features intertwined throughout the forward pass. This intricate architecture affects performance, as seen for the image encoding task discussed above:
>
> | Method | MSE (*10^-4) | SSIM | PSNR | Train Time (s) |
> |-|-|-|-|-|
> |LG-SIREN (1,1)|**12.9**|**0.939**|**32.52 ± 0.66**|87|
> |SIREN|18.4|0.914|31.17 ± 0.68|34|
>
> Additionally, the extreme case of (1,1) is somewhat analogous to INCODE, where a small modulator network augments the features of a large network. It is not surprising that LG-SIREN in this case achieves high reconstruction quality, even though it does not explicitly perform local feature learning. We will make sure that these distinctions are clearly presented in the revised paper.
>
> Regarding why larger partition factors enhances training speed, note that larger partition factors (i.e. *smaller partitions*) results in a *larger* amount of *smaller* local sub-networks, which means there is less interconnectivity between neurons in the entire architecture. This quadratically reduces the number of floating-point operations (FLOPs) in the linear layers, and allows for better parallelization, thereby speeding up training. We briefly mention the quadratic complexity of FLOPs in Section 3.4, but we agree that this point needs to be more explicit. We will make sure it is clear in the final revision of the paper.
>
> Regarding the observation made by [1], it seems like increasing the number of partitions is beneficial in terms of PSNR when using segmentation-based partitions (PoS), but this is not necessarily the case for a fixed grid (PoG) and SIREN, which is similar to our use-case. Also, please note that in our experiments, we explore a significantly larger number of partitions (from 2x2 up to 64x64 in the results above), and in Figure 6 in [1], the authors experiment with up to 12 partitions. It would be interesting to explore the effects of *significantly* increasing the number of partitions for PoS. We once again thank you for pointing our attention towards [1], this work is extremely interesting and relevant to our own.

---

> > ### Comment · Reviewer_XqfY · 2024-08-10
> > **Official Comment from Reviewer XqfY**
> >
> > I appreciate the author's effort in providing such a detailed rebuttal. After reading all the reviewers' opinions and the rebuttal, I think this method showcases that partition-based INR can be used to crop the image in an elegant manner, which is a great potential application of partition-based INR and may contribute to object detection based on INR.
> >
> > Most of my concerns have been well addressed. I tend to raise my rating to vote for "weak accept".

---

> > > ### Author Response · Authors · 2024-08-13
> > >
> > > We thank the reviewer for considering our rebuttal in such detail.
> > > We appreciate the reviewer’s support and the decision to raise the rating.
> > >
> > > The authors

---

### Official Review · Reviewer_AXyi · 2024-07-09

**Soundness:** 4
**Presentation:** 4
**Contribution:** 3
**Rating:** 8
**Confidence:** 5

**Summary:**

The paper proposes a method for learning patchwise INRs that are integrated with a global INR. The method is designed with cropping in mind, and this cropping can be achieved by pruning the relevant patchwise INR - similarly, cropping is limited to the pre-defined patches. This allows for novel post-training cropping, where the INR can be cropped in a way that reduces the number of model weights (and therefore the storage space). The method shows benefits beyond cropping - faster training time (or better visual quality at equal epochs). It is also flexible, and can be applied to various MLP-based INRs.

**Strengths:**

[S1] The paper is exceptionally easy to follow, with good reasoning/motivation, well-explained method, and experiments that support the claims of the paper. Last paragraph of the intro, Figure 1, and Figure 3 are especially good in this regard.

[S2] While on face the applications seem limited, since the pruned parameters can be discarded, this method could be quite useful in the compression regime, where the MLP INR could be cropped by the end user and size would be reduced accordingly. The faster training time is also beneficial in this setting.

[S3] The method has benefits beyond its target problem (cropping) - better quality with less training.

[S4] The experiments are very thorough, proving the flexibility of the method with applications across multiple models and domains.

[S5] The supplementary even goes beyond what's necessary to provide comparisons with e.g. KD for INR training.

**Weaknesses:**

The patch-based formulation is a little unsatisfying in the following ways.

[W1] It doesn't reveal anything very new about the INRs themselves. Rather than discover pruneable parameters that correspond to cropping, or developing some objective that imparts some locality on the representation, the local vs. global distinction is very rigidly enforced as a prior.

[W2] The image can only be cropped according to the pre-defined patches.

Separately,

[W3] Partly connected to W1, the post-training editing is totally restricted to only cropping. This might have applications to compression, but is otherwise at this point more of a novelty, in the sense that it is not very practical.

**Questions:**

What are the practical applications of this method? The main reason my score isn't higher is because I think the impact of this work is somewhat limited to more nice applications that might be of interest to those that study INR, but less so to the broader community.

What does this reveal about the nature of INRs? It would also be nice if the paper revealed some deeper insights about INR in general.

**Limitations:**

Yes.

---

> ### Author Rebuttal · Authors · 2024-08-05
>
> Dear reviewer, thank you for the thoughtful feedback and insightful comments. We appreciate the time and effort. Please see our comments and additional results below.
>
> **Weaknesses:**
>
> **[W1]** We appreciate the reviewer’s observation regarding the rigid enforcement of the local-global distinction as a prior. This rigidity is indeed central to achieving zero-shot croppability in our method. While it may not reveal novel aspects of INRs themselves, our approach demonstrates that (1) combining local and global features serves as a robust prior, enhancing reconstruction quality compared to the INR-per-partition method, (2) the quadratic complexity of fully connected layers in INRs can be relaxed, and (3) there is potential for partition-based downstream tasks and editing.
>
> **[W2]** We agree that the current method only allows cropping according to predefined patches. As mentioned by reviewer *XqfY*, integrating semantic segmentation-based partitioning could address this limitation. Although this is beyond the scope of the current paper, we find it an exciting direction for future research.
>
> **[W3]** We acknowledge that post-training editing is currently limited to cropping. Our long-term vision is to expand this capability, enabling more sophisticated partition-based editing operations. We believe this work lays groundwork for future developments in editable INRs.
>
> **Questions:**
>
> **Practical Applications:** While our research is particularly relevant to those exploring INRs, we believe it also offers broader implications. By extending a baseline MLP-based INR with our local-global approach, we achieve enhanced quality and latency. This improvement can benefit various lines of work incorporating INRs, including image reconstruction, super-resolution, and potentially more complex tasks. In addition, we believe our work can be used for partition-based downstream tasks. For instance, while Functa [13] has shown the use of INRs for classification, our partition-based approach can be used for more complex tasks such as object detection (per-partition classification). Additionally, the locality of weights in our method could facilitate partition-based editing, with methods which directly operate over the weight space such as [29].
>
> **Insights about INRs:** While our work might not provide groundbreaking new insights into the nature of INRs, it demonstrates that integrating local and global features is a promising approach for improving reconstruction quality and reducing computational complexity.
>
> We aim to lay foundations for future partition-based editing and inspire the development of INRs which are inherently editable.
>
> We appreciate the reviewer’s feedback and hope that our comments help illustrate the potential impact and future directions of our work.

---

> > ### Comment · Reviewer_AXyi · 2024-08-12
> > **Rating unchanged**
> >
> > I appreciate the rebuttal. I still think the work is strong, but not groundbreaking, so I keep my rating at strong accept.

---

> ### Author Response · Authors · 2024-08-13
>
> We greatly appreciate the reviewer's support for our work.
>
> The authors

---

### Official Review · Reviewer_45rR · 2024-07-14

**Soundness:** 3
**Presentation:** 4
**Contribution:** 2
**Rating:** 5
**Confidence:** 3

**Summary:**

This paper proposes a new INR architecture to admit easy cropping of the target datum to a certain partition, allowing one to save memory and inference cost without any retraining. Comparing with training a new INR for the target partition, the approach lets one utilize the global context as well. The idea is to train multiple local networks, and modulate their intermediate features with the features of a global network. Through experiments on image/audio/video encoding, the paper demonstrates that the method enjoys faster and more accurate fitting than training one INR per partition. Also, the paper shows that the method works when combined with many INR architectures, e.g., SIREN and INCODE.

**Strengths:**

- A notable strength of the proposed method is that it is very simple and easy to use, making it likely to be scalable and generally applicable. I believe that the method can also be combined well with the triplane-based neural fields or instant-ngp.

- The method is also very clearly presented. In particular, Figure 3 is very effective in delivering how the proposed architecture works.

- Lastly, I appreciate the fact that the paper provides experiments on many different modalities, from image to video.

**Weaknesses:**

- **Utility of croppability?** The key weakness of this paper is the motivation. Apart from "saving storage & compute," the practical utility of having a croppable INR is unclear; will local-global INRs also be useful in performing any subsequent "editing" operations? I suspect that this is why authors provide section 4.4, where the authors "extend" the LGS to parameterize the larger image than the one originally considered. However, for such purposes, there are already other good meta-learning-based solutions such as [26]. I am not sure why the model should be croppable for such applications.

- **Novelty & Ablations.** The proposed method bears much similarity with [26], which modulates the local model with another global model. The key difference here is how we modulate; this paper uses an extra linear layer to process the local+global features, while [26] uses multiplications (later works, such as functa, used addition). To fully understand what this paper contributes, there should be an explicit comparison with these similar methods as a baseline.

- **Evaluation.** If I understood correctly, the evaluation is mostly based on how the model fits the seen coordinates. I wonder how these affect the generalizability of the learned signal to unseen coordinates. As this is one of the key strengths of having a global context, I have enough reasons to believe that LGS will work well. However, an explicit verification is needed.

- **Hyperparameter tuning.** I wonder how the hyperparameters for the models and the baselines are selected. In particular, how were the values of "omega" and the learning rate selected? These two are quite critical in determining the fitting speed, so this point should be crystal clear.

**Questions:**

Please see the "weakness."

**Limitations:**

Yes.

---

> ### Author Rebuttal · Authors · 2024-08-05
>
> Dear reviewer, thank you for the constructive feedback. We appreciate the time and effort. Please see our comments and additional results below.
>
> **Utility of croppability**
>
> The utility of our method stems from the inspiration for editable INRs. We began with the fundamental cropping operation and demonstrated various other benefits. The primary goal is the ability to identify which neural network weights to remove based on the required target signal in a zero-shot manner. Our method offers several additional advantages, as detailed in the paper: **improved accuracy** over the alternative INR-per-partition method and the baseline INR, **faster convergence** in terms of iterations and latency, and **enhanced performance** on various downstream tasks such as CT reconstruction, super-resolution, and denoising. Additionally, we showed its adaptability across multiple modalities.
>
> Regarding more advanced editing operations, we believe our proposed architecture provides a foundation for more sophisticated per-partition editing operations and downstream tasks. Methods that directly manipulate the network’s weight space, such as [29], can leverage the inherent locality of our method.
>
> Regarding section 4.4 - We demonstrate that our method can be used for extending signals. However, it does not compete with other meta-learning solutions. Instead, it serves as a *complementary* method, as one can meta-learn a good initialization for Local-Global INRs and benefit from both. Note that the additional benefit of extending signals with our method lies in the ability to *enlarge the number of weights* to fit larger signals, which is not possible with meta-learning-based solutions that require learning a new initialization. The intuition is demonstrated in Figure 7, where the small SIREN reaches stagnation since the large signal requires more parameters to properly encode.
>
> **Novelty & Ablations**
>
> While the mentioned method bears a resemblance, it does not support cropping of the encoded signal with proportionate decrease in neural network weights, which is the main goal of our architecture. Although [26] improved encoding capabilities with a local-global approach, the network remains a fully-connected INR and thus cannot be modified as discussed. In contrast, our method  leverages an NN architecture that uniformly distributes weights across signal partitions, inherently supports removing unwanted weights, and lays the foundation for future extensions of per-partition editing over the weight space.
>
> On the other hand, one might consider our Local-Global SIREN (LGS) as a potential replacement for [26]’s synthesizer. This idea closely resembles our proposed Local-Global INCODE (LGI). In LGI, we retain the harmonizer (similar to the modulator in [26]) and replace the composer network (similar to the synthesizer in [26]) with a Local-Global version. Since LGI improved both encoding quality and performance on various downstream tasks, we believe replacing the modulator in [26] with a LGS could also enhance the method.
>
> Having said that, we provide a comparison with [26] on the DIV2K subset mentioned in our paper. We trained [26] with the default configuration in the official implementation. Since their implementation currently only supports square images, we used the seven 512x512 samples from the DIV2K subset. The methods were trained from scratch on a single image for 2k iterations, sampling all coordinates in each iteration. Note that [26] requires significantly more training iterations (>60k) to converge, while our LGI achieves better performance after only 2k iterations. *Additionally, [26] has 5.5 times the number of trainable parameters compared to ours.*
>
> | | [26] | SIREN| LGS| INCODE| LGI | [26] (64k iterations) |
> |-|-|-|-|-|-|-|
> |Average PSNR|23.85|32.18|32.88|37.85|**38.37**|35.21|
> |#Params|1.1M|199k|200k|205k|205k|1.1M|
>
> **Evaluation**
>
> We follow the evaluation methods used in many previous INR architecture papers, focusing on various signal encoding capabilities. Towards the end of our paper, we present three downstream tasks where our Local-Global version outperforms the baseline.
> Regarding generalizability, this is a great point that we also addressed in the paper (Section 4.6). Specifically, we train an INCODE and a LGI on a 4x downsampled version of an image and then evaluate the SSIM/PSNR on the full-resolution image. This experiment replicates the one presented in [20]. To provide further evidence of our method's generalizability, we conducted this experiment on the entire DIV2K subset (25 images) mentioned in the paper. The results below show that the Local-Global approach manages to enhance generalization capabilities compared to the baseline INR.
> | Method| Mean PSNR | Mean SSIM |
> |-|-|-|
> |INCODE|26.68|0.703|
> |LGI|**26.76**|**0.734**|
>
> **Hyperparameters**
>
> To ensure a fair comparison, we used the same omega values as in the original SIREN and INCODE implementations. For the LR, we primarily followed the default configurations from baseline methods. In some experiments, we made slight adjustments to the learning rate, ensuring that the selected LR benefits all compared architectures. We will emphasize these points in the final revision of the paper.
>
> For the Local-Global configuration, hyperparameters are selected using our automatic partitioning logic. This logic determines hidden layer dimensions based on a target network size and target global weights ratios (please refer to the general comment for the pseudocode). Detailed experiment configurations and hyperparameters can be found in Appendices A.1 and A.2, as well as in the provided code implementation.

---

> > ### Comment · Reviewer_45rR · 2024-08-12
> >
> > Thank you for the detailed response.
> >
> > **Croppability.** The response makes some sense. However, some points are still unclear to me.
> > - The combinability with meta-learning has not been verified experimentally. In fact, this method seems like one specific way to perform meta-learning.
> > - The argument that "enlarging the number of weights is impossible with meta-learning" is exaggerated; one can simply use multiple models (e.g., the one generated by MAML). Instead, I do agree to the point that meta-learning-based solutions may not be as flexible as the proposed one, having a much larger weight-granularity.
> >
> > **Novelty and ablations.** Thank you for the detailed experiment.
> >
> > **Evaluation.** I may have missed section 4.6. Thank you for pointing this out.
> >
> > **Hyperparameters.** Thank you for stating this.
> >
> > ---
> > Many of my concerns have been verified (with some due to my misunderstanding). Although I am still slightly worried about certain points, I am no longer against acceptance of this paper.

---

> > > ### Author Response · Authors · 2024-08-13
> > >
> > > We thank the reviewer for the detailed response, additional comments, and for raising the score.
> > >
> > > Regarding the additional points:
> > > * We agree that definitive claims regarding the applicability of meta-learning techniques on a Local-Global architecture would require explicit experimental verification. While we have not identified specific limitations that would prevent a Local-Global SIREN network from benefiting from a similar meta-learning approach as applied to SIREN, we acknowledge that further research is needed in this area and plan to explore this in future work.
> > > * We also agree that stating "enlarging the number of weights is impossible with meta-learning" is not entirely accurate. The rephrasing suggested by the reviewer, noting that meta-learning-based solutions are not "as flexible," is indeed more appropriate.
> > >
> > > We appreciate the reviewer’s reconsideration and their support towards the acceptance of our paper.
> > >
> > > The authors

---

### Author Rebuttal · Authors · 2024-08-05

We would like to express our sincere gratitude to all the reviewers for their valuable feedback and insightful comments. We have carefully addressed each point raised in the individual reviews and provided detailed responses in the corresponding review replies.

Additionally, we recognize that the details of the automatic partitioning method were not entirely clear in the initial submission. The automatic partitioning determines the hidden layer dimensions based on the target network size, global weights ratio, and the selected partition size. To clarify, we are providing the pseudocode for the automatic partitioning process below, which will be added to the revised paper:


	FUNCTION AutomaticPartitioning(target_total_params, target_global_weight_ratio, target_partition_size, signal_resolution):
		groups = ComputeNumGroups(signal_resolution, target_partition_size)

		// Compute global hidden dimension
		global_hidden_dim = FindDimension(target_total_params * target_global_weight_ratio)
		global_weights = ComputeGlobalWeights(global_hidden_dim)

		// Compute local hidden dimension
		target_local_weights = target_total_params - global_weights
		local_hidden_dim = FindDimension(target_local_weights, groups)
		local_weights = ComputeLocalWeights(llocal_hidden_dim, groups)

		// Adjust global weights to meet target ratio and total parameter count
		WHILE (global_weights + local_weights) NOT WITHIN 1% OF target_total_params:
			IF (global_weights + local_weights) > target_total_params:
				global_hidden_dim -= 1
			ELSE:
				global_hidden_dim += 1
			global_weights = ComputeGlobalWeights(global_hidden_dim)

		RETURN local_hidden_dim, global_hidden_dim

	// ComputeNumGroups: Calculates the number of partitions the signal is divided to, using simple division
	// ComputeGlobalWeights: Calculates the number of weights in the global network based on the given configuration
	// ComputeLocalWeights: Calculates the number of weights in local networks based on the given configuration
	// FindDimension: Uses binary search to find optimal hidden dimension given target weight count and network parameters.

Thank you,

The authors

---

### Decision · Program_Chairs · 2024-09-25

**Decision:**

Accept (poster)

**Comment:**

The reviewers agreed that the paper is well written and showcases an interesting and effective method to perform cropping.
There were some concerns initially about the motivation and evaluation of the method, but they were addressed adequately and all reviewers are in favour of accepting the paper.